# Noether Theorem and Nilpotency Property of the (Anti-)BRST Charges in the BRST Formalism: A Brief Review

Amit Kumar Rao [1] , Ankur Tripathi [1] , Bhupendra Chauhan [1] and Rudra Prakash Malik [1,2,*]

1 Department of Physics, Institute of Science, Banaras Hindu University, Varanasi 221 005, India
2 DST Centre for Interdisciplinary Mathematical Sciences, Institute of Science, Banaras Hindu University, Varanasi 221 005, India
* Correspondence: malik@bhu.ac.in

**Abstract:** In some of the physically interesting gauge systems, we show that the application of the Noether theorem does *not* lead to the deduction of the Becchi–Rouet–Stora–Tyutin (BRST) and anti-BRST charges that obey *precisely* the off-shell nilpotency property despite the fact that these charges are (*i*) derived by using the off-shell nilpotent (anti-)BRST symmetry transformations, (*ii*) found to be the generators of the *above* continuous symmetry transformations, and (*iii*) conserved with respect to the time-evolution due to the Euler–Lagrange equations of motion derived from the Lagrangians/Lagrangian densities (that describe the dynamics of these suitably chosen physical systems). We propose a *systematic* method for the derivation of the off-shell nilpotent (anti-)BRST charges from the corresponding *non-nilpotent* Noether (anti-)BRST charges. To corroborate the sanctity and preciseness of our proposal, we take into account the examples of (*i*) the one (0 + 1)-dimensional (1D) system of a massive spinning (i.e., SUSY) relativistic particle, (*ii*) the D-dimensional non-Abelian one-form gauge theory, and (*iii*) the Abelian two-form and the St*ü*ckelberg-modified version of the *massive* Abelian three-form gauge theories in any arbitrary D-dimension of spacetime. Our present endeavor is a brief review where some decisive proposals have been made and a few novel results have been obtained as far as the nilpotency property is concerned.

**Keywords:** Noether theorem in BRST approach; continuous (anti-)BRST symmetries; conserved Noether currents; conserved (anti-)BRST charges; nilpotency property

**PACS:** 11.15.-q; 12.20.-m; 03.70.+k

---

## 1. Introduction

It is an undeniable truth that the symmetries of all kinds (e.g., global, local, discrete, continuous, spacetime, internal, etc.) have played a decisive role in the realm of theoretical physics as they have provided a set of deep insights into various working aspects of the physical systems of interest. The basic principles behind the *local* gauge symmetries and diffeomorphism symmetries provide the precise theoretical descriptions of the standard model of particle physics and theory of gravity (i.e., general theory of relativity and (super)string theories). The Becchi–Rouet–Stora–Tyutin (BRST) formalism [1–4] is applied fruitfully to the gauge theories as well as the diffeomorphism invariant theories. Some of the salient features of BRST formalism are (*i*) it covariantly quantizes the gauge theories which are characterized by the existence of the first-class constraints on them in the terminology of Dirac's prescription for the classifications of constraints (see, e.g., [5,6] for details), (*ii*) it is consistent with the Dirac quantization scheme because the physicality criteria with the nilpotent and conserved (anti-)BRST charges imply that the physical states, in the *total* quantum Hilbert space, are *those* that are annihilated by the operator form of the first-class constraints of the gauge theories (see, e.g., [7–9] for details), (*iii*) it maintains the unitarity and *quantum* gauge (i.e., BRST) invariance at any arbitrary order of perturbative

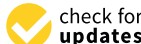



computations of a given physical process that is allowed by an *interacting* gauge theory, and (*iv*) it has deep connections with some of the key ideas behind differential geometry and its (anti-)BRST transformations resemble the $\mathcal{N} = 2$ supersymmetry transformations. There is a decisive difference, however, between the key properties associated with the (anti-)BRST symmetries and the $\mathcal{N} = 2$ supersymmetries in the sense that the *nilpotent* BRST and anti-BRST symmetries are absolutely anticommuting in nature *but* the *nilpotent* $\mathcal{N} = 2$ supersymmetries are *not*. To sum up, we note that the application of the BRST formalism is physically very useful and mathematically, its horizon is quite wide.

The purpose of our present endeavor is related with the derivations of the conserved (anti-)BRST charges by exploiting the theoretical potential of Noether's theorem and a thorough study of *their* nilpotency property. We take into account diverse examples of physically interesting models of gauge theories and demonstrate that the Noether theorem does not always lead to the derivation of conserved and *nilpotent* (anti-)BRST charges[1]. One has to apply specific set of theoretical tricks and techniques to obtain the *nilpotent* versions of the (anti-)BRST charges from the Noether conserved (anti-)BRST charges which are found to be *non-nilpotent*. In simple examples, we show that one equation of motion is good enough to convert the non-nilpotent Noether conserved (anti-)BRST charges into the conserved and *nilpotent* (anti-)BRST charges. For instance, in the case of a 1D *massive* spinning (i.e., SUSY) relativistic particle (see, e.g., [10–13] and references therein), the Euler–Lagrange (EL) equations of motion (EoM) with respect to the "gauge" and "supergauge" variables are sufficient to convert a non-nilpotent set of (anti-)BRST charges into *nilpotent* of order two (cf. Section 2). However, in the case of a D-dimensional non-Abelian 1-form theory, the Gauss divergence theorem and EL-EoM with respect to the gauge field are needed to obtain the *nilpotent* (anti-)BRST charges from the conserved and non-nilpotent Noether (anti-)BRST charges. We have purposely chosen these simple examples so that it becomes clear that the celebrated Noether theorem does *not* lead to the derivation of nilpotent (anti-)BRST charges where (*i*) the non-trivial (anti-)BRST invariant Curci–Ferrari (CF) type restrictions exist, and (*ii*) a set of coupled (but equivalent) Lagrangians/Lagrangian densities respects the off-shell nilpotent (anti-)BRST symmetry transformations.

It is worthwhile to mention here that the limiting cases of the above *two* examples are the free scalar relativistic particle and D-dimensional Abelian 1-form gauge theory where there is the existence of a *single* Lagrangian/Lagrangian density. In these cases, the Noether conserved (anti-)BRST charges are off-shell nilpotent automatically because the CF-type restriction is *trivial*. In fact, it is observed that the *trivial* CF-type restriction of the scalar relativistic particle is the limiting case of the non-trivial CF-type restriction of the spinning relativistic particle and the *trivial* CF-type restriction of the Abelian 1-form gauge theory is the limiting case of the *non-trivial* CF-condition of non-Abelian 1-form gauge theory.

One of the central issue we address in our present investigation is the cases of the BRST approach to higher $p$-form ($p = 2, 3, \ldots$) gauge theories[2] where there is always the existence of (*i*) a set of coupled (but equivalent) Lagrangian densities and (*ii*) a set of (anti-)BRST invariant CF-type restrictions. In such cases, the Noether theorem always leads to the derivation of conserved (anti-)BRST charges which are found to be *non-nilpotent*. In fact, the expressions for these charges are quite complicated, and the EL-EoMs are too many because of the presence of too many fields in the theory (cf. Section 5 below for details). We make a *systematic* proposal which enables us to obtain the off-shell nilpotent (anti-) BRST charges from the conserved Noether (anti-)BRST charges which are found to be non-nilpotent. One of the key ingredients of our proposal is the observation that one has to start with the EL-EoM with respect to the gauge field of the $p$-form gauge theories where, most of the time, the Gauss divergence theorem is required to be applied (before we exploit the potential and power of the EL-EoM with respect to the gauge fields). To be precise, all the D-dimensional ($D \geq 2$) higher $p$-form gauge theories require the application of the Gauss divergence theorem before we exploit the potential of EL-EoM with respect to the gauge field. After this, it is the interplay amongst (*i*) the application of the EL-EoMs, (*ii*) use of the (anti-) BRST transformations, and (*iii*) the requirement of Gauss's divergence

theorem that lead to the derivation of the nilpotent versions of the (anti-)BRST charges from the conserved Noether (anti-)BRST charges which are found to be non-nilpotent (cf. Sections 4–6).

The theoretical contents of our present endeavor are organized as follows. In Section 2, we exploit the theoretical potential and power of Noether's theorem in the context of a gauge system of 1D spinning relativistic particles to deduce the explicit expressions for the conserved (anti-)BRST charges $Q_{(a)b}$ and establish that they are *not* off-shell nilpotent. We focus, in Section 3, on the D-dimensional non-Abelian 1-form gauge theory (without any interaction with matter fields) and demonstrate that the Noether conserved (anti-)BRST charges, once again, are non-nilpotent to begin with. We pinpoint the *specific* EL-EoMs that have to be used to make these (anti-)BRST charges off-shell nilpotent. Section 4 is devoted to the discussion of Noether's theorem in the context of D-dimensional (anti-)BRST invariant Abelian 2-form theory and discuss the nitty-gritty details of the nilpotency property. The theoretical content of Section 5 is concerned with the (anti-)BRST invariant coupled (but equivalent) Lagrangian densities of the (anti-)BRST invariant *modified* massive Abelian three-form gauge theory, and our discussion is centered around the property of the off-shell nilpotency of the (anti-)BRST charges. Finally, in Section 6, we make some concluding remarks and comment on the future prospects of our present investigation.

In our Appendix A, we discuss the Stückelberg-modified D-dimensional massive Abelian two-form theory and deduce the off-shell nilpotent versions of the (anti-)BRST charges $[Q^{(1)}_{(a)b}]$ from the non-nilpotent Noether conserved (anti-)BRST charges $[Q_{(a)b}]$.

*Convention and Notations for the 1D Massive Spinning (SUSY) Relativistic Particle and D-dimensional Abelian Two-Form as well as Three-Form Theories*: We follow the convention of the left derivatives with respect to all the fermionic variables/fields of our theory in the computations of the canonical conjugate momenta and the Noether conserved currents. The flat metric tensor $\eta_{\mu\nu} = \text{diag}\,(+1, -1, -1, \dots)$ is chosen for the D-dimensional *flat* Minkowskian space so that the dot product between two non-null vectors $P_\mu$ and $Q_\mu$ is denoted by $P \cdot Q = \eta_{\mu\nu} P^\mu Q^\nu = P_0 Q_0 - P_i Q_i$ where the Greek indices $\mu, \nu, \lambda, \dots = 0, 1, 2, \dots D - 1$ correspond to the time and space directions and the Latin indices $i, j, k \dots = 1, 2, \dots D - 1$ stand for the space directions *only*. Throughout the whole body of our text, the nilpotent (anti-)BRST symmetry transformations carry the symbol $s_{(a)b}$, and the corresponding conserved (anti-) BRST charges are denoted by $Q_{(a)b}$ for *all* kinds of the Abelian systems that have been chosen for our present discussion. For our discussions on the D-dimensional non-Abelian gauge theory, we shall adopt a different convention in Section 3.

## 2. Preliminary: (Anti-)BRST Charges and Nilpotency for a Massive Spinning Relativistic Particle

We begin with the following coupled (but equivalent) (anti-)BRST invariant (see, e.g., [12,13]) Lagrangians that describe the dynamics of a one $(0 + 1)$-dimensional *massive* spinning (i.e., supersymmetric) relativistic particle in the D-dimensional target space, namely:

$$
\begin{aligned}
L_b \;=\; & L_f + b^2 + b\,(\dot{e} + 2\,\bar{\beta}\,\beta) - i\,\dot{\bar{c}}\,\dot{c} + \bar{\beta}^2\,\beta^2 + 2\,i\,\chi\,(\beta\,\dot{\bar{c}} - \bar{\beta}\,\dot{c}) - 2\,e\,(\bar{\beta}\,\dot{\beta} + \gamma\,\chi) \\
& +\; 2\,\gamma\,(\beta\,\bar{c} - \bar{\beta}\,c) + m\,(\bar{\beta}\,\dot{\beta} - \dot{\bar{\beta}}\,\beta + \gamma\,\chi) - \dot{\gamma}\,\psi_5,
\end{aligned}
\tag{1}
$$

$$
\begin{aligned}
L_{\bar{b}} \;=\; & L_f + \bar{b}^2 - \bar{b}\,(\dot{e} - 2\,\bar{\beta}\,\beta) - i\,\dot{\bar{c}}\,\dot{c} + \bar{\beta}^2\,\beta^2 + 2\,i\,\chi\,(\beta\,\dot{\bar{c}} - \bar{\beta}\,\dot{c}) + 2\,e\,(\dot{\bar{\beta}}\,\beta - \gamma\,\chi) \\
& +\; 2\,\gamma\,(\beta\,\bar{c} - \bar{\beta}\,c) + m\,(\bar{\beta}\,\dot{\beta} - \dot{\bar{\beta}}\,\beta + \gamma\,\chi) - \dot{\gamma}\,\psi_5,
\end{aligned}
\tag{2}
$$

where $L_f$ is the first-order Lagrangian for our system [10]

$$
L_f = p_\mu\,\dot{x}^\mu + \frac{i}{2}\,(\psi_\mu\,\dot{\psi}^\mu - \psi_5\,\dot{\psi}_5) - \frac{e}{2}\,(p^2 - m^2) + i\,\chi\,(p_\mu\,\psi^\mu - m\,\psi_5).
\tag{3}
$$

In the above, the target space canonical conjugate quantities $(x_\mu(\tau), p^\mu(\tau))$ are the bosonic coordinates $(x_\mu)$ and canonical momenta $(p^\mu)$, respectively, with the Greek indices $\mu = 0, 1, 2, \ldots, D-1$ corresponding to the D-dimensional *flat* Minkowskian target space. The trajectory of the spinning particle is parameterized by $\tau$ and the generalized velocities: $\dot{x}_\mu = (d\,x^\mu/d\,\tau)$, $\dot{\psi}_\mu = (d\,\psi^\mu/d\,\tau)$ is defined with respect to *it*. The pair of fermionic ($\psi_\mu^2 = 0$; $\psi_\mu \psi_\nu + \psi_\nu \psi_\mu = 0$, $\psi_5^2 = 0$, $\psi_5 \psi_\mu + \psi_\mu \psi_5 = 0$, etc.) variables $(\psi_\mu, \psi_5)$ are introduced in the theory to (i) maintain the SUSY gauge symmetry transformations and (ii) incorporate the *mass*-shell condition $p^2 - m^2 = 0$ where $m$ is the rest mass of the particle. The variable $e(\tau)$ and $\chi(\tau)$ are the superpartners of each other where $e(\tau)$ is the einbein variable and the fermionic ($\chi^2 = 0$) variable $\chi(\tau)$ is its superpartner and *both* of them behave as the "gauge" and "supergauge" variables. We have incorporated a pair of variables $(b, \bar{b})$ as the *bosonic* Nakanishi–Lautrup type auxiliary variables which participate in defining the CF-type restriction[3]: $b + \bar{b} + 2\,\bar{\beta}\,\beta = 0$ where the *bosonic* (anti-)ghost variables $(\bar{\beta})\,\beta$ are the counterparts of the *fermionic* ($c^2 = 0$, $\bar{c}^2 = 0$, $c\,\bar{c} + \bar{c}\,c = 0$) (anti-)ghost variables $(\bar{c})\,c$ in our supersymmetric (i.e., spinning) system of a 1D diffeomorphism invariant theory [10,12,13]. We require an additional auxiliary variable $(\gamma)$ that is *fermionic* $(\gamma^2 = 0)$ in nature as it anticommutes with *all* the other *fermionic* variables of our theory (i.e., $\gamma\chi + \chi\gamma = 0$, $\gamma\psi_5 + \psi_5\gamma = 0$, $\gamma\psi_\mu + \psi_\mu\gamma = 0$, $\gamma c + c\gamma = 0$, $\gamma\bar{c} + \bar{c}\gamma = 0$, etc.).

The above Lagrangians $L_b$ and $L_{\bar{b}}$ are coupled (but equivalent) on the submanifold of the *quantum* variables where the CF-type restriction: $b + \bar{b} + 2\,\bar{\beta}\,\beta = 0$ is satisfied [12,13]. Furthermore, it is straightforward to check that under the following off-shell nilpotent $[s_{(a)b}^2 = 0]$ (anti-)BRST symmetry transformations $[s_{(a)b}]$, namely;

$$s_{ab}\,x_\mu = \bar{c}\,p_\mu + \bar{\beta}\,\psi_\mu, \qquad s_{ab}\,e = \dot{\bar{c}} + 2\,\bar{\beta}\,\chi, \qquad s_{ab}\,\psi_\mu = i\,\bar{\beta}\,p_\mu,$$

$$s_{ab}\,\bar{c} = -i\,\bar{\beta}^2, \quad s_{ab}\,c = i\,\bar{b}, \quad s_{ab}\,\bar{\beta} = 0, \quad s_{ab}\,\beta = -i\,\gamma, \quad s_{ab}\,p_\mu = 0,$$

$$s_{ab}\,\gamma = 0, \quad s_{ab}\,\bar{b} = 0, \quad s_{ab}\,\chi = i\,\dot{\bar{\beta}}, \quad s_{ab}\,b = 2\,i\,\bar{\beta}\,\gamma, \quad s_{ab}\,\psi_5 = i\,\bar{\beta}\,m, \qquad (4)$$

$$s_b\,x_\mu = c\,p_\mu + \beta\,\psi_\mu, \qquad s_b\,e = \dot{c} + 2\,\beta\,\chi, \qquad s_b\,\psi_\mu = i\,\beta\,p_\mu,$$

$$s_b\,c = -i\,\beta^2, \quad s_b\,\bar{c} = i\,b, \quad s_b\,\beta = 0, \quad s_b\,\bar{\beta} = i\,\gamma, \quad s_b\,p_\mu = 0,$$

$$s_b\,\gamma = 0, \quad s_b\,b = 0, \quad s_b\,\chi = i\,\dot{\beta}, \quad s_b\,\bar{b} = -2\,i\,\beta\,\gamma, \quad s_b\,\psi_5 = i\,\beta\,m, \qquad (5)$$

the Lagrangians $L_b$ and $L_{\bar{b}}$ transform[4] to the *total* derivatives as:

$$s_{ab}\,L_{\bar{b}} = \frac{d}{d\,\tau}\left[\frac{\bar{c}}{2}\,(p^2 + m^2) + \frac{\bar{\beta}}{2}\,(p_\mu\,\psi^\mu + m\,\psi_5) - \bar{b}\,(\dot{\bar{c}} + 2\,\bar{\beta}\,\chi)\right], \qquad (6)$$

$$s_b\,L_b = \frac{d}{d\,\tau}\left[\frac{c}{2}\,(p^2 + m^2) + \frac{\beta}{2}\,(p_\mu\,\psi^\mu + m\,\psi_5) + b\,(\dot{c} + 2\,\beta\,\chi)\right]. \qquad (7)$$

As a consequence, the action integrals $S_1 = \int_{-\infty}^{\infty} d\,\tau\,L_b$ and $S_2 = \int_{-\infty}^{\infty} d\,\tau\,L_{\bar{b}}$ remain invariant for the *physically* well-defined variables that vanish off as $\tau \to \pm\infty$. The application of Noether's theorem yields the following explicit expressions for the conserved (anti-)BRST charges $[Q_{(a)b}]$ for our 1D SUSY system of a relativistic spinning particle, namely:

$$Q_{ab} = \frac{\bar{c}}{2}\,(p^2 - m^2) + \bar{\beta}\,(p_\mu\,\psi^\mu - m\,\psi_5) - \bar{b}\,\dot{\bar{c}} - 2\,\bar{b}\,\bar{\beta}\,\chi - i\,m\,\bar{\beta}\,\gamma - \bar{\beta}^2\,\dot{c} - 2\,\beta\,\bar{\beta}^2\,\chi, \quad (8)$$

$$Q_b = \frac{c}{2}\,(p^2 - m^2) + \beta\,(p_\mu\,\psi^\mu - m\,\psi_5) + b\,\dot{c} + 2\,b\,\beta\,\chi - i\,m\,\beta\,\gamma + \beta^2\,\dot{\bar{c}} + 2\,\bar{\beta}\,\beta^2\,\chi, \quad (9)$$

where we have used the following explicit relationships

$$
\begin{aligned}
Q_{ab} &= (s_{ab}\, x_\mu)\, \frac{\partial L_{\bar{b}}}{\partial \dot{x}_\mu} + (s_{ab}\, \psi_\mu)\, \frac{\partial L_{\bar{b}}}{\partial \dot{\psi}_\mu} + (s_{ab}\, e)\, \frac{\partial L_{\bar{b}}}{\partial \dot{e}} + (s_{ab}\, c)\, \frac{\partial L_{\bar{b}}}{\partial \dot{c}} + (s_{ab}\, \bar{c})\, \frac{\partial L_{\bar{b}}}{\partial \dot{\bar{c}}} \\
&\quad + (s_{ab}\, \beta)\, \frac{\partial L_{\bar{b}}}{\partial \dot{\beta}} + (s_{ab}\, \psi_5)\, \frac{\partial L_{\bar{b}}}{\partial \dot{\psi}_5} - X,
\end{aligned}
\tag{10}
$$

$$
\begin{aligned}
Q_{b} &= (s_{b}\, x_\mu)\, \frac{\partial L_{b}}{\partial \dot{x}_\mu} + (s_{b}\, \psi_\mu)\, \frac{\partial L_{b}}{\partial \dot{\psi}_\mu} + (s_{b}\, e)\, \frac{\partial L_{b}}{\partial \dot{e}} + (s_{b}\, c)\, \frac{\partial L_{b}}{\partial \dot{c}} + (s_{b}\, \bar{c})\, \frac{\partial L_{b}}{\partial \dot{\bar{c}}} \\
&\quad + (s_{b}\, \bar{\beta})\, \frac{\partial L_{b}}{\partial \dot{\bar{\beta}}} + (s_{b}\, \psi_5)\, \frac{\partial L_{b}}{\partial \dot{\psi}_5} - Y,
\end{aligned}
\tag{11}
$$

for the derivations of conserved (anti-)BRST charges $Q_{(a)b}$.

The following points are pertinent as far as Equations (10) and (11) are concerned. First, even though some of the auxiliary variables (e.g., $b$, $\bar{b}$, $\chi$) transform (cf. Equations (4) and (5)) under the (anti-)BRST symmetry transformations, we have *not* used their contributions because their "time" derivative is *not* present in the Lagrangians $L_b$ and $L_{\bar{b}}$. Second, we have *not* taken into account the contributions of $\beta$ and $\bar{\beta}$ in (11) and (10), respectively, because $s_{ab}\,\bar{\beta} = 0$ and $s_b\,\bar{\beta} = 0$. Third, the expressions X and Y are the quantities that are present in the square brackets of (6) and (7). Fourth, the direct applications of the EL-EoMs ensure that $Q_{(a)b}$ are conserved quantities [13]. Finally, the conserved charges $Q_{(a)b}$ are the generators for the *continuous* (anti-)BRST symmetry transformations (4) and (5) because it can be checked that the following is *true*, namely:

$$
s_r\, \Phi = -i\, [\Phi, Q_r]_\pm, \qquad r = b,\, ab,
\tag{12}
$$

where the $(\pm)$ signs (as the subscript) on the square bracket, on the r.h.s., denote that the square bracket is an (anti)commutator for the given variable $\Phi$ being fermionic/bosonic in nature. Here, $\Phi$ denotes the generic variable of $L_b$ and $L_{\bar{b}}$. In other words, we have: $\Phi = x_\mu,\, \phi_\mu,\, \psi_\mu,\, \psi_5,\, b,\, \bar{b},\, e,\, \chi,\, c,\, \bar{c},\, \beta,\, \bar{\beta},\, \gamma$. Using the principle behind the continuous symmetries and their corresponding generators (cf. Equation (12)), we have the following

$$
s_b\, Q_b = -i\, \{Q_b,\, Q_b\}, \qquad s_{ab}\, Q_{ab} = -i\, \{Q_{ab},\, Q_{ab}\},
\tag{13}
$$

where the l.h.s. of *both* the entries can be *directly* computed by using the (anti-)BRST symmetry transformations (cf. Equations (4) and (5)) and the explicit expressions for the Noether conserved (anti-)BRST charges (8) and (9). It is interesting to point out that the explicit computations of $s_b\, Q_b$ and $s_{ab}\, Q_{ab}$ are as follows:

$$
\begin{aligned}
s_b\, Q_b &= i\, \beta^2 \left[ \dot{b} + \frac{1}{2}\, (p^2 - m^2) + 2\, \gamma\, \chi + 2\, \bar{\beta}\dot{\beta} \right] \equiv -i\, \{Q_b,\, Q_b\}, \\
s_{ab}\, Q_{ab} &= i\, \bar{\beta}^2 \left[ \frac{1}{2}\, (p^2 - m^2) - \dot{b} + 2\, \gamma\, \chi - 2\, \beta\, \dot{\bar{\beta}} \right] \equiv -i\, \{Q_{ab},\, Q_{ab}\}.
\end{aligned}
\tag{14}
$$

It is obvious, from the above expressions, that the (anti-)BRST charges $Q_{(a)b}$ are *not* off-shell nilpotent. However, these *non-nilpotent* conserved charges can be made off-shell nilpotent if $(i)$ we use the EL-EoMs with respect to the "gauge" and "supergauge" variables $e\,(\tau)$ and $\chi\,(\tau)$, respectively, which are, in some sense, superpartners of each other, $(ii)$ we apply the principle that the off-shell nilpotent *continuous* (anti-)BRST transformations are generated by the conserved (anti-)BRST charges, and $(iii)$ we apply the (anti-)BRST symmetry transformations at appropriate places.

We propose here a *systematic* method to obtain the off-shell *nilpotent* version $Q_b^{(1)}$ of the BRST charge $Q_b$. Our aim would be to obtain $Q_b^{(1)}$ from the *non-nilpotent* Noether BRST charge from $Q_b$ such that $s_b\, Q_b^{(1)} = -i\, \{Q_b^{(1)},\, Q_b^{(1)}\} = 0$. In other words, the l.h.s. (i.e., $s_b\, Q_b^{(1)}$) should be *precisely* equal to zero. Toward this goal in mind, first of all, we focus on

the EL-EoMs that emerge out from $L_b$ with respect to $e\,(\tau)$ and $\chi\,(\tau)$ which are the "gauge" and "supergauge" variables. These, in their useful form, are as follows:

$$\frac{1}{2}\,(p^2 - m^2) = -\dot{b} - 2\,(\bar{\beta}\,\dot{\beta} + \gamma\,\chi),$$
$$(p_\mu\,\psi^\mu - m\,\psi_5) = -i\,m\,\gamma + 2\,i\,e\,\gamma - 2\,(\beta\,\dot{\bar{c}} - \bar{\beta}\,\dot{c}). \tag{15}$$

In the *second* step, the substitutions of EL-EoMs with respect to "gauge" and "super-gauge" variables in the appropriate terms of Noether conserved BRST charge $Q_b$. For instance, the substitutions of (15) lead to the modifications of the following terms:

$$\frac{c}{2}\,(p^2 - m^2) = -\dot{b}\,c - 2\,c\,(\bar{\beta}\,\dot{\beta} + \gamma\,\chi),$$
$$\beta\,(p_\mu\,\psi^\mu - m\,\psi_5) = -i\,m\,\beta\,\gamma + 2\,i\,e\,\beta\,\gamma - 2\,\beta\,(\beta\,\dot{\bar{c}} - \bar{\beta}\,\dot{c}). \tag{16}$$

In the *third* step, we observe whether the above "modified" terms add, subtract and/or cancel out with some of the terms of $Q_b$. For instance, we note that in our present case, only the term which is added in (16) is "$-i\,m\,\beta\,\gamma$" from $Q_b$. The total sum of the expressions in (16) and *this* term is the following explicit expression:

$$-\dot{b}\,c - 2\,c\,(\bar{\beta}\,\dot{\beta} + \gamma\,\chi) - 2\,i\,m\,\beta\,\gamma + 2\,i\,e\,\beta\,\gamma - 2\,\beta\,(\beta\,\dot{\bar{c}} - \bar{\beta}\dot{c}). \tag{17}$$

In the *fourth* step, we apply the BRST transformations on (17) which yields:

$$-i\,b\,\beta^2 + 2\,i\,\beta^2\,\chi\,\gamma - 2\,i\,\beta^2\,\dot{\beta}\,\bar{\beta}. \tag{18}$$

In our *fifth* step, we keenly observe whether some of terms of the Noether conserved charge $Q_b$ should be *modified* so that the terms of (18) cancel out *precisely when* we apply the BRST symmetry transformations on *them*. In our present case, we observe *luckily* that

$$s_b\,[\beta^2\,\dot{\bar{c}} + 2\,\beta^2\,\bar{\beta}\,\chi] = i\,b\,\beta^2 + 2\,i\,\beta^2\,\gamma\,\chi + 2\,i\,\beta^2\,\dot{\beta}\,\bar{\beta}, \tag{19}$$

which cancels out whatever we have obtained in (18). In the *final* step, we apply the BRST symmetry transformations on the leftover terms of the Noether conserved charge $Q_b$. It turns out that we have the following

$$s_b\,[b\,\dot{c} + 2\,b\,\beta\,\chi] = -2\,i\,b\,\beta\,\dot{\beta} + 2\,i\,b\,\beta\,\dot{\beta} = 0. \tag{20}$$

It is pertinent to point out that all the terms that cancel out due to the application of the BRST symmetry transformations should be *present* in the off-shell nilpotent version of the (anti-)BRST charges $Q^{(1)}_{(a)b}$. For instance, *all* the terms of (17) and the terms, on the l.h.s. of (19) and (20) in the square bracket, will be present[5] in $Q^{(1)}_b$. Ultimately, in our present case, we obtain the following off-shell nilpotent version of the BRST charge:

$$Q_b \longrightarrow Q^{(1)}_b = b\,\dot{c} - \dot{b}\,c + 2\,\beta\,[i\,e\,\gamma + \bar{\beta}\,\dot{c} - i\,m\,\gamma + b\,\chi + \beta\,\bar{\beta}\,\chi] - \beta^2\dot{\bar{c}} - 2\,c\,[\bar{\beta}\,\dot{\beta} + \gamma\,\chi]. \tag{21}$$

At this stage, it is straightforward to note that the following observation is *true*, namely:

$$s_b\,Q^{(1)}_b = -i\,\{Q^{(1)}_b,\,Q^{(1)}_b\}, \quad \Longrightarrow \quad [Q^{(1)}_b]^2 = 0. \tag{22}$$

In other words, we point out that the off-shell *nilpotent* version of the conserved BRST charge $[Q^{(1)}_b]$ is obtained from the *non-nilpotent* Noether conserved charge by using the EL-EoMs with respect to the "gauge" variable $e\,(\tau)$ and "supergauge" variable $\chi\,(\tau)$ and the application of the (anti-)BRST symmetry transformations at appropriate places.

Against the backdrop of the above paragraph, we note that to obtain the off-shell nilpotent version of the conserved anti-BRST charge, we use the following EL-EoMs

$$\frac{1}{2}\,(p^2 - m^2) = \dot{\bar{b}} + 2\,(\dot{\bar{\beta}}\,\beta - \gamma\,\chi),$$
$$(p_\mu\,\psi^\mu - m\,\psi_5) = 2\,i\,e\,\gamma - i\,m\,\gamma - 2\,(\beta\,\dot{\bar{c}} - \bar{\beta}\,\dot{c}). \tag{23}$$

that are derived from the *perfectly* anti-BRST invariant $L_{\bar{b}}$. We follow *exactly* the same steps as in the case of BRST charge $Q_b$ to obtain the off-shell nilpotent version of the anti-BRST $Q_{ab}^{(1)}$. In fact, the substitution of (23) into the expression for $Q_{ab}$ (cf. Equation (8)) at appropriate places leads to the following expression for $Q_{ab}^{(1)}$, namely:

$$Q_{ab} \longrightarrow Q_{ab}^{(1)} = \dot{\bar{b}}\,\bar{c} - \bar{b}\,\dot{c} + 2\,\bar{\beta}\,[i\,e\,\gamma - \beta\,\dot{\bar{c}} - \bar{b}\,\chi - i\,m\,\gamma - \beta\,\bar{\beta}\,\chi]$$
$$+ \; 2\,\bar{c}\,(\bar{\beta}\,\dot{\beta} + \gamma\,\chi) + \bar{\beta}^2\,\dot{c}. \tag{24}$$

It is now straightforward to note that we have the following:

$$s_{ab}\,Q_{ab}^{(1)} = -\,i\,\{Q_{ab}^{(1)},\,Q_{ab}^{(1)}\}, \quad \Longrightarrow \quad [Q_{ab}^{(1)}]^2 = 0. \tag{25}$$

The above observation is nothing but the proof of the off-shell nilpotency of the anti-BRST charge $Q_{ab}^{(1)}$ where the l.h.s. is computed explicitly by using (4) and (24).

We end this section with the following remarks. First of all, we note that it is the EL-EoMs with respect to the "gauge" and "supergauge" variables that have been used and these have been *singled out* from the *rest* of the EL-EoM. This observation is one of the key ingredients of our *proposal* followed by the steps that have been discussed from Equation (15) to Equation (20). Second, in our present *simple* case of a 1D spinning relativistic particle, only a *single* step is good enough to enable us to obtain an off-shell nilpotent set of (anti-)BRST conserved charges. However, we shall see that in the context of Abelian two-form and three-form gauge theories defined in any arbitrary dimension of spacetime, *more* steps will be required to obtain the off-shell nilpotent set of conserved (anti-)BRST charges from the *non-nilpotent* forms of the (anti)BRST charges (that are derived *directly* from the applications of Noether's theorem). Third, it is interesting to mention that in the limiting case of the spinning relativistic particle when $\beta = \bar{\beta} = \gamma = 0$, we obtain the (anti-)BRST charges for the *scalar* relativistic particle from (8) and (9) as

$$Q_{ab} \longrightarrow Q_{ab}^{(sr)} = \frac{\bar{c}}{2}\,(p^2 - m^2) - \bar{b}\,\dot{c}, \qquad Q_b \longrightarrow Q_b^{(sr)} = \frac{c}{2}\,(p^2 - m^2) + b\,\dot{c}, \tag{26}$$

which are off-shell nilpotent of order two because $s_b\,Q_b^{(sr)} = 0$ and $s_{ab}\,Q_{ab}^{(sr)} = 0$ due to: $s_b\,c = 0,\,s_b\,b = 0,\,s_b\,p_\mu = 0$ *and* $s_{ab}\,\bar{c} = 0,\,s_{ab}\,\bar{b} = 0,\,s_{ab}\,p_\mu = 0$ which are *true* for the scalar relativistic particle. In the above equation, the superscript $(sr)$ on the charges stand for the conserved and off-shell nilpotent (anti-)BRST charges for the *scalar* relativistic particle. Finally, it can be explicitly checked that the *modified* versions of the (anti-)BRST charges $Q_{(a)b}^{(1)}$ are *also* conserved quantities if we use the *proper* EL-EoMs that are derived from the coupled (but equivalent) Lagrangians $L_b$ and $L_{\bar{b}}$ of our 1D *massive* SUSY gauge theory.

## 3. (Anti-)BRST Charges and Nilpotency: Arbitrary Dimensional Non-Abelian 1-Form Gauge Theory

In this section, we show that the *Noether* conserved (anti-)BRST charges for the D-dimensional non-Abelian one-form gauge theory are *non-nilpotent*. However, following our proposal, we can obtain the appropriate forms of the conserved and off-shell nilpotent expressions for the (anti-)BRST charges for our non-Abelian theory (without any inter-

actions with the matter fields). We begin with the following coupled (but equivalent) Lagrangian densities (see, e.g., [9]) in the Curci–Ferrari gauge (see, e.g., [15,16])

$$\mathcal{L}_B = -\frac{1}{4} F^{\mu\nu} \cdot F_{\mu\nu} + B \cdot (\partial_\mu A^\mu) + \frac{1}{2} (B \cdot B + \bar{B} \cdot \bar{B}) - i \partial_\mu \bar{C} \cdot D^\mu C,$$

$$\mathcal{L}_{\bar{B}} = -\frac{1}{4} F^{\mu\nu} \cdot F_{\mu\nu} - \bar{B} \cdot (\partial_\mu A^\mu) + \frac{1}{2} (B \cdot B + \bar{B} \cdot \bar{B}) - i D^\mu \bar{C} \cdot \partial_\mu C, \tag{27}$$

where the field strength tensor $F_{\mu\nu} \equiv F_{\mu\nu}^a T^a$ ($a = 1, 2, \ldots, N^2 - 1$) has been derived from the non-Abelian two-form: $F^{(2)} = d A^{(1)} + i A^{(1)} \wedge A^{(1)}$ where the one-form $A^{(1)} = dx^\mu A_\mu \equiv d x^\mu A_\mu^q T^a$ defines the non-Abelian gauge field ($A_\mu^a$) so that we have: $F_{\mu\nu}^a = \partial_\mu A_\nu^a - \partial_\nu A_\mu^a + i f^{abc} A_\mu^b A_\nu^c$. For the $SU(N)$ Lie algebraic space, we have the Lie algebra: $[T^a, T^b] = f^{abc} T^c$ that is satisfied by the $SU(N)$ generators $T^a$ ($a = 1, 2, \ldots, N^2 - 1$) where $f^{abc}$ are the structure constants that can be chosen [17] to be *totally* antisymmetric in *all* the indices for the semi-simple Lie group $SU(N)$. In this section, we adopt the dot and cross products *only* in the $SU(N)$ Lie algebraic space where we have: $S \cdot T = S^a T^a$, $(S \times T)^a = f^{abc} S^b T^c$ for the two non-null vectors $S^a$ and $T^a$ (in *this* space and $a, b, c, \cdots = 1, 2, \ldots N^2 - 1$). We have also taken into account the summation convention where the repeated indices are summed over and $D_\mu C = \partial_\mu C + i (A_\mu \times C)$ and $D_\mu \bar{C} = \partial_\mu \bar{C} + i (A_\mu \times \bar{C})$ are the covariant derivatives in the adjoint representation of the $SU(N)$ Lie algebra.

The above coupled (but equivalent) Lagrangian densities (27) respect the following off-shell nilpotent [$s_{(a)b}^2 = 0$] (anti-)BRST symmetry transformations ($s_{(a)b}$)

$$s_{ab} A_\mu = D_\mu \bar{C}, \qquad s_{ab} \bar{C} = -\frac{i}{2} (\bar{C} \times \bar{C}), \qquad s_{ab} C = i\bar{B}, \qquad s_{ab} \bar{B} = 0,$$

$$s_{ab} F_{\mu\nu} = i (F_{\mu\nu} \times \bar{C}), \qquad s_{ab}(\partial_\mu A^\mu) = \partial_\mu D^\mu \bar{C}, \qquad s_{ab} B = i (B \times \bar{C}),$$

$$s_b A_\mu = D_\mu C, \qquad s_b C = -\frac{i}{2} (C \times C), \qquad s_b \bar{C} = i B, \qquad s_b B = 0,$$

$$s_b \bar{B} = i (\bar{B} \times C), \qquad s_b (\partial_\mu A^\mu) = \partial_\mu D^\mu C, \qquad s_b F_{\mu\nu} = i (F_{\mu\nu} \times C), \tag{28}$$

because of the following observations

$$s_b \mathcal{L}_B = \partial_\mu (B \cdot D^\mu C), \quad s_{ab} \mathcal{L}_{\bar{B}} = -\partial_\mu (\bar{B} \cdot D^\mu \bar{C}),$$

$$s_b \mathcal{L}_{\bar{B}} = \partial_\mu [\{B + (C \times \bar{C})\} \cdot \partial^\mu C] - \{B + \bar{B} + (C \times \bar{C})\} \cdot D_\mu \partial^\mu C,$$

$$s_{ab} \mathcal{L}_B = -\partial_\mu [\{\bar{B} + (C \times \bar{C})\} \cdot \partial^\mu \bar{C}] + \{(B + \bar{B} + (C \times \bar{C})\} \cdot D_\mu \partial^\mu \bar{C}, \tag{29}$$

which establish that the action integrals $S_1 = \int d^D x \, \mathcal{L}_B, S_2 = \int d^D x \, \mathcal{L}_{\bar{B}}$ remain invariant ($s_b S_1 = 0$, $s_{ab} S_2 = 0$) under the BRST and anti-BRST symmetry transformations, respectively, because the *physical* fields vanish off as $x \longrightarrow \pm\infty$ due to Gauss's divergence theorem. If we confine our whole discussion on the submanifold of the *total* quantum Hilbert space of fields where the CF-condition: $B + \bar{B} + (C \times \bar{C}) = 0$ is respected [18], we note that *both* the Lagrangian densities respect *both* the nilpotent symmetries. In other words, we have $s_b \mathcal{L}_{\bar{B}} = -\partial_\mu [\bar{B} \cdot \partial^\mu C]$, $s_{ab} \mathcal{L}_B = \partial_\mu [B \cdot \partial^\mu \bar{C}]$ on the *above* submanifold of Hilbert space of quantum fields. We christen the transformations: $s_b \mathcal{L}_B = \partial_\mu [B \cdot D^\mu C]$, $s_{ab} \mathcal{L}_{\bar{B}} = -\partial_\mu [\bar{B} \cdot D^\mu \bar{C}]$ as *perfect* symmetry transformations because we do *not* use any EL-EoMs and/or CF-type condition for *their* proof.

In addition to the *equivalence* of the *coupled* Lagrangian densities (27) from the point of view of the (anti-)BRST symmetry considerations, we note that the absolute anticommutativity (i.e., $\{s_b, s_{ab}\} = 0$) property of the (anti-)BRST symmetry transformations is satisfied if and only if we invoke the sanctity of the CF-condition: $B + \bar{B} + (C \times \bar{C}) = 0$. This becomes obvious when we observe that the following are *true*, namely:

$$\{s_b, s_{ab}\} A_\mu = i D_\mu [B + \bar{B} + (C \times \bar{C})],$$

$$\{s_b, s_{ab}\} F_{\mu\nu} = -F_{\mu\nu} \times [B + \bar{B} + (C \times \bar{C})]. \tag{30}$$

Thus, it is clear that the absolute anticommutativity properties: $\{s_b, s_{ab}\} A_\mu = 0$ and $\{s_b, s_{ab}\} F_{\mu\nu} = 0$ are *true* if and only if $B + \bar{B} + (C \times \bar{C}) = 0$. We further note that $\{s_b, s_{ab}\} \Phi = 0$ where $\Phi = C, \bar{C}, B, \bar{B}$ is the generic field of the theory (besides $A_\mu$ and $F_{\mu\nu}$). Thus, the absolute anticommutativity property (i.e., $\{s_b, s_{ab}\} \Phi = 0$) is *automatically* satisfied for the fields: $B, \bar{B}, C, \bar{C}$ due to the off-shell nilpotent (anti-)BRST symmetry transformations (28). It is very interesting to point out that the straightforward *equivalence* $(\mathcal{L}_B = \mathcal{L}_{\bar{B}})$ of *both* the Lagrangian densities (27) of our theory leads to

$$(\partial_\mu A^\mu) \cdot \left[ B + \bar{B} + (C \times \bar{C}) \right] = 0, \tag{31}$$

modulo a total spacetime derivative. The above observation establishes the fact that *both* the Lagrangian densities $\mathcal{L}_B$ and $\mathcal{L}_{\bar{B}}$ of Equation (27) are *coupled* in the sense that the Nakanishi–Lautrup auxiliary fields $B$ and $\bar{B}$ are *not* free but *these* specific fields are restricted to obey $B + \bar{B} + (C \times \bar{C}) = 0$ (which is nothing but the CF condition [18]). This condition, for the $SU(N)$ non-Abelian gauge theory, is *physically* sacrosanct because it is an (anti-)BRST invariant (i.e., $s_{(a)b} [B + \bar{B} + (C \times \bar{C})] = 0$) quantity, which can be verified by using the (anti-)BRST symmetry transformations (28).

The *perfect* symmetry invariance of the Lagrangian densities $\mathcal{L}_B$ and $\mathcal{L}_{\bar{B}}$ under the infinitesimal, continuous and off-shell nilpotent $[s_{(a)b}^2 = 0]$ BRST and anti-BRST symmetry transformations, respectively, leads to the derivation of the following expressions for the conserved Noether currents

$$J_{(B)}^\mu = (s_b A_\nu) \frac{\partial \mathcal{L}_B}{\partial (\partial_\mu A_\nu)} + (s_b \bar{C}) \frac{\partial \mathcal{L}_B}{\partial (\partial_\mu \bar{C})} + (s_b C) \frac{\partial \mathcal{L}_B}{\partial (\partial_\mu C)} - B \cdot D^\mu C,$$

$$J_{(\bar{B})}^\mu = (s_{ab} A_\nu) \frac{\partial \mathcal{L}_{\bar{B}}}{\partial (\partial_\mu A_\nu)} + (s_{ab} C) \frac{\partial \mathcal{L}_{\bar{B}}}{\partial (\partial_\mu C)} + (s_{ab} \bar{C}) \frac{\partial \mathcal{L}_{\bar{B}}}{\partial (\partial_\mu \bar{C})} + \bar{B} \cdot D^\mu \bar{C}, \tag{32}$$

where we have followed the convention of the left derivative with respect to all the *fermionic* fields. The explicit form of the (anti-)BRST Noether currents are:

$$J_{(\bar{B})}^\mu = - F^{\mu\nu} \cdot D_\nu \bar{C} - \bar{B} \cdot D^\mu \bar{C} - \frac{1}{2} (\bar{C} \times \bar{C}) \cdot \partial^\mu C,$$

$$J_{(B)}^\mu = B \cdot D^\mu C - F^{\mu\nu} \cdot D_\nu C + \frac{1}{2} \partial^\mu \bar{C} \cdot (C \times C). \tag{33}$$

The conservation law $\partial_\mu J_{(r)}^\mu = 0$ (with $r = \bar{B}, B$) can be proven by exploiting the power and potential of the EL-EoMs. For the proof of $\partial_\mu J_{(\bar{B})}^\mu = 0$, we have to use the following EL-EOMs with respect to the gauge field $A_\mu$ and (anti-)ghost fields $(\bar{C}) C$, namely:

$$D_\mu F^{\mu\nu} + \partial^\nu \bar{B} + (\bar{C} \times \partial^\nu C) = 0, \qquad \partial_\mu (D^\mu \bar{C}) = 0, \qquad D_\mu (\partial^\mu C) = 0, \tag{34}$$

that are derived from $\mathcal{L}_{\bar{B}}$. In exactly similar fashion, for the proof of $\partial_\mu J_{(B)}^\mu = 0$, we utilize the following EL-EoMs with respect to $A_\mu$, $C$ and $\bar{C}$, namely:

$$D_\mu F^{\mu\nu} - \partial^\nu B - (\partial^\nu \bar{C} \times C) = 0, \qquad \partial_\mu (D^\mu C) = 0, \qquad D_\mu (\partial^\mu \bar{C}) = 0, \tag{35}$$

which are derived from the Lagrangian density $\mathcal{L}_B$. Ultimately, we claim that the Noether currents (33) are *conserved*, and they lead to the derivation of *conserved* (anti-)BRST charges for our D-dimensional non-Abelian one-form gauge theory. Following the sacrosanct prescription of Noether theorem, we derive the expressions for the *conserved* (anti-)BRST charges $(Q_{(\bar{B})B} = \int d^{D-1}x \, J_{(\bar{B})B}^0)$ as follows:

$$Q_{\bar{B}} = - \int d^{D-1}x \left[ F^{0i} \cdot D_i \bar{C} + \bar{B} \cdot D_0 \bar{C} + \frac{1}{2} (\bar{C} \times \bar{C}) \cdot \dot{C} \right],$$

$$Q_B = \int d^{D-1}x \left[ B \cdot D_0 C - F^{0i} \cdot D_i C + \frac{1}{2} \dot{\bar{C}} \cdot (C \times C) \right]. \tag{36}$$

A few comments, at this juncture, are in order. First of all, it can be checked that $\partial_0 Q_{\bar{B}} = 0$ and $\partial_0 Q_B = 0$ where we have to use the EL-EoMs from $\mathcal{L}_B$ and $\mathcal{L}_{\bar{B}}$, and (ii) the above conserved (anti-)BRST charges are the generators of *all* the symmetry transformations (28) provided we use the canonical (anti-)commutators by deriving the explicit expressions for the canonical conjugate momenta from $\mathcal{L}_B$ and $\mathcal{L}_{\bar{B}}$. Using the principle behind the continuous symmetry transformations and their generators as the Noether conserved charges, we note that the expressions (36) lead to the following

$$s_{ab} Q_{\bar{B}} = -\int d^{D-1}x \left[ i \left( F^{0i} \times \bar{C} \right) \cdot D_i \bar{C} - \frac{i}{2} \left( \bar{C} \times \bar{C} \right) \cdot \dot{\bar{B}} \right] \neq 0,$$

$$s_b Q_B = \int d^{D-1}x \left[ -i \left( F^{0i} \times C \right) \cdot D_i C + \frac{i}{2} \dot{B} \cdot \left( C \times C \right) \right] \neq 0, \qquad (37)$$

which are *not* equal to zero. In other words, we note that: $s_{ab} Q_{\bar{B}} = -i \{Q_{\bar{B}}, Q_{\bar{B}}\} \neq 0$ and $s_b Q_B = -i \{Q_B, Q_B\} \neq 0$. Hence, the expressions for the Noether (anti-)BRST charges (36) are *not* off-shell nilpotent (i.e., $Q_{\bar{B}}^2 \neq 0$, $Q_B^2 \neq 0$) of order *two*.

Following the *proposal* mentioned in the context of 1D *massive* spinning relativistic particle, in the first step, we have to find out the EL-EOMs with respect to the non-Abelian gauge field and substitute it in the *non-nilpotent* Noether conserved charges $Q_B$ and $Q_{\bar{B}}$. In our present case, it can be done *only* after the *application* of the Gauss divergence theorem so that we have the following for the *first* term in $Q_{\bar{B}}$ and the *second* term $Q_B$, namely:

$$\int d^{D-1}x \left[ -F^{0i} \cdot \left( \partial_i \bar{C} + i A_i \times \bar{C} \right) \right] \equiv +\int d^{D-1}x \left( \partial_i F^{0i} \right) \cdot \bar{C}$$
$$- i \int d^{D-1}x \left[ F^{0i} \cdot \left( A_i \times \bar{C} \right) \right], \qquad (38)$$

$$\int d^{D-1}x \left[ -F^{0i} \cdot \left( \partial_i C + i A_i \times C \right) \right] = +\int d^{D-1}x \left( \partial_i F^{0i} \right) \cdot C$$
$$- i \int d^{D-1}x \left[ F^{0i} \cdot \left( A_i \times C \right) \right], \qquad (39)$$

where we can use the following EL-EoM with respect to the gauge field, namely:

$$\left( \partial_i F^{0i} \right) \cdot \bar{C} = \dot{\bar{B}} \cdot \bar{C} + \left( \bar{C} \times \dot{C} \right) \cdot \bar{C} - i \left( A_i \times F^{0i} \right) \cdot \bar{C},$$
$$\left( \partial_i F^{0i} \right) \cdot C = -\dot{B} \cdot C + \left( \dot{\bar{C}} \times C \right) \cdot C - i \left( A_i \times F^{0i} \right) \cdot C. \qquad (40)$$

In the *second* step, we have to find out if there are addition, subtraction and/or cancellations with the *rest* of the terms of the conserved Noether charges $Q_{\bar{B}}$ and $Q_B$ in (36). At this stage, taking the help of: $+(\bar{C} \times \dot{C}) \cdot \bar{C} = +(\bar{C} \times \bar{C}) \cdot \dot{C}$ and $-(\dot{\bar{C}} \times C) \cdot C = -\dot{\bar{C}} \cdot (C \times C)$, we have the following expressions for $Q_{(\bar{B})B}^{(1)}$ from the expressions for $Q_{\bar{B}}$ and $Q_B$, namely:

$$Q_{\bar{B}} \to Q_{\bar{B}}^{(1)} = \int d^{D-1}x \left[ \dot{\bar{B}} \cdot \bar{C} - \bar{B} \cdot D_0 \bar{C} + \frac{1}{2} \left( \bar{C} \times \bar{C} \right) \cdot \dot{C} \right],$$

$$Q_B \to Q_B^{(1)} = \int d^{D-1}x \left[ B \cdot D_0 C - \dot{B} \cdot C - \frac{1}{2} \dot{\bar{C}} \cdot \left( C \times C \right) \right], \qquad (41)$$

where (*i*) a cancellation has taken place between $\left[ -i \left( A_i \times F^{0i} \right) \cdot \bar{C} \right]$ and $\left[ -i F^{0i} \cdot \left( A_i \times \bar{C} \right) \right]$ in the expression for $Q_{\bar{B}}$, (*ii*) in the expression for $Q_B$, there has been a cancellation between $\left[ -i \left( A_i \times F^{0i} \right) \cdot C \right]$ and $\left[ -i F^{0i} \cdot \left( A_i \times C \right) \right]$, (*iii*) the pure ghost terms have been added to yield $\left[ +\frac{1}{2} \left( \bar{C} \times \bar{C} \right) \cdot \dot{C} \right]$ in $Q_{\bar{B}}$ and $-\frac{1}{2} \dot{\bar{C}} \cdot \left( C \times C \right)$ in the expression for $Q_B$, and (*iv*) the *final* contributions, after the applications of the Gauss divergence theorem (cf. Equations (38) and (39)) and the equation of motion (40), are as follows

$$\dot{\bar{B}} \cdot \bar{C} + \frac{1}{2} \left( \bar{C} \times \bar{C} \right) \cdot \dot{C}, \qquad -\dot{B} \cdot C - \frac{1}{2} \dot{\bar{C}} \cdot \left( C \times C \right), \qquad (42)$$

in $Q_{\bar{B}}$ and $Q_B$, respectively. In the *third* step, we have to apply the anti-BRST and BRST symmetry transformations on (42), respectively. It turns out that we have the following:

$$s_{ab}\left[\dot{\bar{B}}\cdot\bar{C}+\frac{1}{2}\,(\bar{C}\times\bar{C})\cdot\dot{C}\right]=0, \qquad s_b\left[-\dot{B}\cdot C-\frac{1}{2}\,\dot{C}\cdot(C\times C)\right]=0. \qquad (43)$$

Thus, our *all* the relevant steps *terminate* here, and we have the *final* expression for the (anti-)BRST charges as quoted in (41) because it is elementary to check that the leftover terms of (36) are (anti-)BRST invariant: $s_{ab}\,(\bar{B}\cdot D_0\,\bar{C})=0$, $s_b\,(B\cdot D_0\,C)=0$. It is straightforward now to observe that the above expressions for the conserved (anti-)BRST charges are off-shell nilpotent $[(Q^{(1)}_{(\bar{B})B})^2=0]$ of order two. To corroborate this statement, we have the following observations related with the *modified* version of $Q^{(1)}_{(\bar{B})B}$

$$\begin{aligned}
s_{ab}\,Q^{(1)}_{\bar{B}}&=-i\,\{Q^{(1)}_{\bar{B}},\,Q^{(1)}_{\bar{B}}\}=0 \quad\Longrightarrow\quad [Q^{(1)}_{\bar{B}}]^2=0,\\
s_b\,Q^{(1)}_{B}&=-i\,\{Q^{(1)}_{B},\,Q^{(1)}_{B}\}=0 \quad\Longrightarrow\quad [Q^{(1)}_{B}]^2=0,
\end{aligned} \qquad (44)$$

where the l.h.s. of the above equation can be computed explicitly by taking the help of (28) and (34). Thus, the (anti-)BRST charges $Q^{(1)}_{(\bar{B})B}$ are off-shell nilpotent.

We end this section with the following *useful* remarks. First, we observe that the Noether conserved (anti-)BRST charges $Q_{\bar{B}}$ and $Q_B$ are *not* off-shell nilpotent of order *two* as was the case with the *massive* spinning (i.e., SUSY) relativistic particle. Second, to obtain the conserved and off-shell nilpotent expressions for the (anti-)BRST charges $Q^{(1)}_{\bar{B}}$ and $Q^{(1)}_B$, we have, first of all, taken the help of Gauss's divergence theorem and, then, used *only* the equation of motion for the *gauge* field from the *coupled* Lagrangian densities $\mathcal{L}_{\bar{B}}$ and $\mathcal{L}_B$. This observation is *similar* to our observation in the context of the 1D *massive* spinning relativistic particle (modulo Gauss's divergence theorem). Finally, we observe that the Abelian *limit* (i.e., $f^{abc}=0$ *plus* no dot and/or cross products *plus* no covariant derivative, etc.) of the Noether conserved charges $Q_{\bar{B}}$ and $Q_B$ from (31) are:

$$\begin{aligned}
Q_{\bar{B}}&\longrightarrow Q_{ab}=-\int d^{D-1}x\left[F^{0i}\,\partial_i\,\bar{C}-B\,\dot{\bar{C}}\right],\\
Q_B&\longrightarrow Q_b=\int d^{D-1}x\left[F^{0i}\,\partial_i\,C-B\,\dot{C}\right],
\end{aligned} \qquad (45)$$

where $Q_{(a)b}$ are the (anti-)BRST charges for the *free* Abelian 1-form gauge theory. It is an elementary exercise to note that the above expressions for the charges are off-shell nilpotent $(s_b\,Q_b=-i\,\{Q_b,\,Q_b\}=0,\ s_{ab}\,Q_{ab}=-i\,\{Q_{ab},\,Q_{ab}\}=0)$ where we have to apply the analogues of the (anti-)BRST transformations (28) on $Q_{(a)b}$ for the Abelian one-form theory theory, which are: $s_b\,F_{0i}=0$, $s_b\,C=0$, $s_b\,B=0$ and $s_{ab}\,F_{0i}=0$, $s_{ab}\,\bar{C}=0$, $s_{ab}\,\bar{B}=0$.

## 4. (Anti-)BRST Charges and Nilpotency: Arbitrary Dimensional Abelian 2-Form Gauge Theory

We begin with the (anti-)BRST invariant coupled (but equivalent) Lagrangian densities for the *free* D-dimensional Abelian two-form gauge theory as follows (see, e.g., [8,19] for details)

$$\begin{aligned}
\mathcal{L}_B&=\frac{1}{12}\,H^{\mu\nu\kappa}\,H_{\mu\nu\kappa}+B^\mu\left(\partial^\nu B_{\nu\mu}-\partial_\mu\phi\right)+B\cdot B+\partial_\mu\bar{\beta}\,\partial^\mu\beta\\
&\quad+\ (\partial_\mu\bar{C}_\nu-\partial_\nu\bar{C}_\mu)\,(\partial^\mu C^\nu)+(\partial\cdot C-\lambda)\,\rho+(\partial\cdot\bar{C}+\rho)\,\lambda,
\end{aligned} \qquad (46)$$

$$\begin{aligned}
\mathcal{L}_{\bar{B}}&=\frac{1}{12}\,H^{\mu\nu\kappa}\,H_{\mu\nu\kappa}+\bar{B}^\mu\left(\partial^\nu B_{\nu\mu}+\partial_\mu\phi\right)+\bar{B}\cdot\bar{B}+\partial_\mu\bar{\beta}\,\partial^\mu\beta\\
&\quad+\ (\partial_\mu\bar{C}_\nu-\partial_\nu\bar{C}_\mu)\,(\partial^\mu C^\nu)+(\partial\cdot C-\lambda)\,\rho+(\partial\cdot\bar{C}+\rho)\,\lambda,
\end{aligned} \qquad (47)$$

where the *totally* antisymmetric tensor $H_{\mu\nu\lambda}=\partial_\mu\,B_{\nu\lambda}+\partial_\nu\,B_{\lambda\mu}+\partial_\lambda\,B_{\mu\nu}$ is derived from the Abelian three-form $H^{(3)}=d\,B^{(2)}\equiv\left[(d\,x^\mu\wedge d\,x^\nu\wedge d\,x^\lambda)/3!\right]H_{\mu\nu\lambda}$. Here, the Abelian two-

form $B^{(2)} = [(d\,x^\mu \wedge d\,x^\nu)/2!]\,B_{\mu\nu}$ is an antisymmetric $(B_{\mu\nu} = -B_{\nu\mu})$ tensor gauge field and $d = d\,x^\mu\,\partial_\mu$ (with $d^2 = 0$) is the exterior derivative. The gauge-fixing term for the gauge field has its origin in the co-exterior derivative of differential geometry (see, e.g., [20–23]), as it is straightforward to check that $\delta\,B^{(2)} = - * d * B^{(2)} \equiv (\partial^\nu\,B_{\nu\mu})\,d\,x^\mu$ where $*$ is the Hodge duality operator on the *flat* D-dimesnional spectime manifold. A derivative on a scalar field $(\phi)$ has been incorporated into the gauge-fixing term on dimensional ground. The Nakanishi–Lautrup-type auxiliary vector fields $B_\mu$ and $\bar B_\mu$ are restricted to obey the CF-type restriction: $B_\mu - \bar B_\mu - \partial_\mu\,\phi = 0$ (see, e.g., [24]). Here, the fermionic ($C_\mu^2 = 0$, $\bar C_\mu^2 = 0$, $C_\mu\,C_\nu + C_\nu\,C_\mu = 0$, $\bar C_\mu\,\bar C_\nu + \bar C_\nu\,\bar C_\mu = 0$, $C_\mu\,\bar C_\nu + \bar C_\nu\,C_\mu = 0$, etc.) vector (anti-)ghost fields $(\bar C_\mu)\,C_\mu$ carry the ghost numbers $(-1) + 1$, respectively, and the bosonic (anti-)ghost fields $(\bar\beta)\,\beta$ are endowed with the ghost numbers $(-2) + 2$, respectively. The auxiliary (anti-)ghost fields $(\rho)\,\lambda$ are fermionic ($\rho^2 = \lambda^2 = 0$, $\rho\,\lambda + \lambda\,\rho = 0$, etc.) in nature, and they *also* carry the ghost numbers $(-1) + 1$, respectively, due to the fact that $\lambda = \frac12\,(\partial \cdot C)$ and $\rho = -\frac12\,(\partial \cdot \bar C)$. The (anti-)ghost fields are invoked to maintain the *unitarity* in the theory.

The above *coupled* Lagrangian densities $\mathcal{L}_B$ and $\mathcal{L}_{\bar B}$ respect the following *perfect* off-shell nilpotent $[s_{(a)b}^2 = 0]$ (anti-)BRST symmetry transformations $[s_{(a)b}]$, namely:

$$
\begin{aligned}
&s_{ab}B_{\mu\nu} = -(\partial_\mu\bar C_\nu - \partial_\nu\bar C_\mu), \quad s_{ab}\bar C_\mu = -\partial_\mu\bar\beta, \qquad s_{ab}C_\mu = \bar B_\mu,\\
&s_{ab}\phi = \rho, \quad s_{ab}\beta = -\lambda, \; s_{ab}B_\mu = \partial_\mu\rho, \quad s_{ab}\big[\rho,\lambda,\bar\beta,\bar B_\mu,H_{\mu\nu\kappa}\big] = 0,\\
&s_b B_{\mu\nu} = -(\partial_\mu C_\nu - \partial_\nu C_\mu), \qquad s_b C_\mu = -\partial_\mu\beta, \qquad s_b\bar C_\mu = -B_\mu,\\
&s_b\phi = \lambda, \quad s_b\bar\beta = -\rho, \; s_b\bar B_\mu = -\partial_\mu\lambda, \quad s_b\big[\rho,\lambda,\beta,B_\mu,H_{\mu\nu\kappa}\big] = 0,
\end{aligned}
\tag{48}
$$

due to our observations that:

$$
\begin{aligned}
s_{ab}\mathcal{L}_{\bar B} &= -\partial_\mu\Big[(\partial^\mu\bar C^\nu - \partial^\nu\bar C^\mu)\bar B_\nu - \rho\,\bar B^\mu + \lambda\partial^\mu\bar\beta\Big],\\
s_b\mathcal{L}_B &= -\partial_\mu\Big[(\partial^\mu C^\nu - \partial^\nu C^\mu)B_\nu + \rho\,\partial^\mu\beta + \lambda B^\mu\Big].
\end{aligned}
\tag{49}
$$

Thus, it is crystal clear that the action integrals $S_1 = \int d^D x\,\mathcal{L}_B$ and $S_2 = \int d^D x\,\mathcal{L}_{\bar B}$ remain invariant (i.e., $s_b S_1 = 0$, $s_{ab}S_2 = 0$) for the *physical* fields that vanish-off as $x \longrightarrow \pm\infty$ due to Gauss's divergence theorem. It should be noted that the above (anti-)BRST symmetry transformations are absolutely anticommuting *only* on the submanifold of the Hilbert space of quantum fields where the CF-type restriction: $B_\mu - \bar B_\mu - \partial_\mu\phi = 0$ is satisfied. This statement can be corroborated by the following observation

$$
\{s_b, s_{ab}\}\,B_{\mu\nu} = \partial_\mu(B_\nu - \bar B_\nu) - \partial_\nu(B_\mu - \bar B_\mu),
\tag{50}
$$

which establishes that $\{s_b, s_{ab}\}\,B_{\mu\nu} = 0$ if and only if we invoke the sanctity of CF-type restriction $B_\mu - \bar B_\mu - \partial_\mu\phi = 0$. It can be checked that the absolute anticommutativity property $\{s_b, s_{ab}\}\,\Psi = 0$ is satisfied *automatically* if we use (48) for the generic field $\Psi = C_\mu, \bar C_\mu, \phi, \lambda, \rho, \beta, \bar\beta, B_\mu, \bar B_\mu$ of $\mathcal{L}_B$ and $\mathcal{L}_{\bar B}$.

The above infinitesimal, continuous and off-shell nilpotent (anti-)BRST symmetry transformations lead to the derivations of the Noether conserved currents

$$
\begin{aligned}
J_{(ab)}^\mu &= \rho\,\bar B^\mu - (\partial^\mu\bar C^\nu - \partial^\nu\bar C^\mu)\,\bar B_\nu - (\partial^\mu C^\nu - \partial^\nu C^\mu)\,\partial_\nu\bar\beta - \lambda\,\partial^\mu\bar\beta - H^{\mu\nu\kappa}\,(\partial_\nu\bar C_\kappa),\\
J_{(b)}^\mu &= (\partial^\mu\bar C^\nu - \partial^\nu\bar C^\mu)\,\partial_\nu\beta - (\partial^\mu C^\nu - \partial^\nu C^\mu)\,B_\nu - \lambda\,B^\mu - \rho\,\partial^\mu\beta - H^{\mu\nu\kappa}\,(\partial_\nu C_\kappa),
\end{aligned}
\tag{51}
$$

where it is quite straightforward to check (see, e.g., [19] for details) that the conservation law $(\partial_\mu J_{(r)}^\mu,\ r = ab,\ b)$ is *true* provided we use the EL-EoMs from the coupled (but equivalent) (anti-)BRST invariant Lagrangian densities $\mathcal{L}_B$ and $\mathcal{L}_{\bar B}$, respectively. The conserved Noether (anti-)BRST charges $Q_r = \int d^{D-1} x\,J_{(r)}^0$ $(r = ab,\ b)$ are as follows

$$Q_{ab} = \int d^{D-1}x \left[ \rho \, \bar{B}^0 - (\partial^0 \bar{C}^i - \partial^i \bar{C}^0) \, \bar{B}_i - (\partial^0 C^i - \partial^i C^0) \, \partial_i \bar{\beta} - \lambda \, \partial^0 \bar{\beta} - H^{0ij} \, (\partial_i \bar{C}_j) \right],$$

$$Q_b = \int d^{D-1}x \left[ (\partial^0 \bar{C}^i - \partial^i \bar{C}^0) \, \partial_i \beta - (\partial^0 C^i - \partial^i C^0) B_i - \lambda \, B^0 - \rho \, \partial^0 \beta - H^{0ij} \, (\partial_i C_j) \right], \quad (52)$$

which are the generators for the infinitesimal, continuous and off-shell nilpotent $[s_{(a)b}^2 = 0]$ (anti-)BRST symmetry transformations (48). At this stage, it is worthwhile to point out that the following observations are *true*, namely:

$$s_{ab} \, Q_{ab} = -i \{ Q_{ab}, Q_{ab} \} = \int d^{D-1}x \left[ -(\partial^0 \bar{B}^i - \partial^i \bar{B}^0) \, \partial_i \bar{\beta} \right] \neq 0,$$

$$s_b \, Q_b = -i \{ Q_b, Q_b \} = \int d^{D-1}x \left[ -(\partial^0 B^i - \partial^i B^0) \, \partial_i \beta \right] \neq 0, \quad (53)$$

when we apply the principle behind the continuous symmetry transformations and their generators as the *conserved* Noether charges. The above observations establish that the Noether conserved charges $Q_b$ and $Q_{ab}$ are *not* off-shell nilpotent (i.e., $Q_b^2 \neq 0$, $Q_{ab}^2 \neq 0$).

At this juncture, we follow our *proposal* to obtain the off-shell nilpotent versions $[Q_{(a)b}^{(1)}]$ of the (anti-)BRST conserved charges from the conserved Noether (anti-)BRST charge $Q_{(a)b}$ [cf. Equation (52)] which are found to be *non-nilpotent* [cf. Equation (53)]. Our objective is to prove the validity of $s_{(a)b} Q_{(a)b}^{(1)} = -i \{ Q_{(a)b}^{(1)}, Q_{(a)b}^{(1)} \} = 0$ by computing precisely the l.h.s. (i.e., $s_{(a)b} Q_{(a)b}^{(1)}$). In the *first* step, we have to use the equation of motion with respect to the gauge field. Toward this goal in mind, first of all, we note that the *last* terms of $Q_b$ and $Q_{ab}$ can be re-expressed as follows

$$- \int d^{D-1}x \, H^{0ij} \, \partial_i C_j = - \int d^{D-1}x \, \partial_i \, [H^{0ij} \, C_j] + \int d^{D-1}x \, (\partial_i H^{0ij}) \, C_j,$$

$$- \int d^{D-1}x \, H^{0ij} \, \partial_i \bar{C}_j = - \int d^{D-1}x \, \partial_i \, [H^{0ij} \, \bar{C}_j] + \int d^{D-1}x \, (\partial_i H^{0ij}) \, \bar{C}_j, \quad (54)$$

where the first-term will be zero due to Gauss's divergence theorem for the physical fields (which vanish-off as $x \longrightarrow \mp \infty$), and we can apply the *first* step of our proposal where the equations of motion with respect to the gauge field from $\mathcal{L}_B$ and $\mathcal{L}_{\bar{B}}$ are as follows:

$$\partial_i H^{0ij} = (\partial^0 B^j - \partial^j B^0), \qquad \partial_i H^{0ij} = (\partial^0 \bar{B}^j - \partial^j \bar{B}^0). \quad (55)$$

Thus, the *last* terms of $Q_{ab}$ and $Q_b$ are as follows:

$$\int d^{D-1}x \, (\partial_i H^{0ij}) \, \bar{C}_j = \int d^{D-1}x \, [(\partial^0 \bar{B}^i - \partial^i \bar{B}^0) \, \bar{C}_i],$$

$$\int d^{D-1}x \, (\partial_i H^{0ij}) \, C_j = \int d^{D-1}x \, [(\partial^0 B^i - \partial^i B^0) \, C_i]. \quad (56)$$

We remark here that the terms in (56) here emerged out from the EL-EoMs with respect to the gauge field, and they are sacrosanct. As a consequence, they will be present in the off-shell nilpotent version of $Q_{(a)b}^{(1)}$. In the *second* step, we apply the (anti-)BRST symmetry transformations on the above terms (modulo integration) which lead to the following:

$$s_{ab} \, [(\partial^0 \bar{B}^i - \partial^i \bar{B}^0) \, \bar{C}_i] = -(\partial^0 \bar{B}^i - \partial^i \bar{B}^0) \, \partial_i \bar{\beta},$$

$$s_b \, [(\partial^0 B^i - \partial^i B^0) \, C_i] = -(\partial^0 B^i - \partial^i B^0) \, \partial_i \beta. \quad (57)$$

In the *third* step, we have to modify[6] appropriate terms of $Q_{ab}$ and $Q_b$ so that (57) cancels out when we apply the (anti-)BRST symmetry transformations on *them*. With this goal in mind, first of all, we focus on the BRST charge $Q_b$ and explain *clearly* the *third* step of our proposal. It can be seen that the *first* term of $Q_b$ can be re-written as

$$(\partial^0 \bar{C}^i - \partial^i \bar{C}^0) \, \partial_i \beta = 2 \, (\partial^0 \bar{C}^i - \partial^i \bar{C}^0) \, \partial_i \beta - (\partial^0 \bar{C}^i - \partial^i \bar{C}^0) \, \partial_i \beta. \quad (58)$$

It is clear that if we apply the BRST symmetry transformations on the *second* term of the above equation, it will serve our purpose and *cancel out* the second entry in (57). At this stage, we comment that the *second* term of (58) and (56) will be *present* in the off-shell nilpotent version $(Q_b^{(1)})$ of the BRST charge. This is due to the fact that our central aim is to show that $s_b Q_b^{(1)} = 0$. Now, we focus on the *first* term of (57), which can be re-written in an appropriate form:

$$2 \left(\partial^0 \bar{C}^i - \partial^i \bar{C}^0\right) \partial_i \beta = \partial_i \left[2 \left(\partial^0 \bar{C}^i - \partial^i \bar{C}^0\right) \beta\right] - 2 \partial_i \left[\partial^0 \bar{C}^i - \partial^i \bar{C}^0\right] \beta. \tag{59}$$

The above terms are inside the integration with respect to $d^{D-1}x$. Hence, the *first* term of the above equation will vanish due to the Gauss divergence theorem. The second term can be written, using the following equations of motion, derived from $\mathcal{L}_B$, as:

$$\Box \bar{C}_\mu - \partial_\mu \left(\partial \cdot \bar{C}\right) - \partial_\mu \rho = 0,$$
$$\implies -2 \partial_i \left[\partial^0 \bar{C}^i - \partial^i \bar{C}^0\right] \beta = 2 \beta \partial^0 \rho \equiv 2 \beta \dot{\rho}. \tag{60}$$

In the *fourth* step, we have to apply the BRST symmetry transformations on $(2 \beta \dot{\rho})$ which turns out to be zero. We comment that *this* term will be present in the off-shell nilpotent version $(Q_b^{(1)})$ of the BRST charge due to our objective to show that $s_b Q_b^{(1)} = 0$. Hence, *all* the steps of our proposal *terminate* at this stage. As a consequence, we have the following off-shell nilpotent version of the BRST charge:

$$\begin{aligned} Q_b \longrightarrow Q_b^{(1)} &= \int d^{D-1}x \left[\left(\partial^0 B^i - \partial^i B^0\right) C_i - \left(\partial^0 \bar{C}^i - \partial^i \bar{C}^0\right) \partial_i \beta \right. \\ &+ \left. 2 \beta \dot{\rho} - \left(\partial^0 C^i - \partial^i C^0\right) B_i - \lambda B^0 - \rho \dot{\beta}\right], \end{aligned} \tag{61}$$

where we have taken into account the appropriate term from (56), the *second* term from (58) and the r.h.s. of (60). It is straightforward now to check that

$$s_b Q_b^{(1)} = -i \left\{Q_b^{(1)}, Q_b^{(1)}\right\} = 0 \implies [Q_b^{(1)}]^2 = 0, \tag{62}$$

which proves the off-shell nilpotency of the *modified* version (i.e., $Q_b^{(1)}$) of the *non-nilpotency* Noether conserved charges $Q_b$. It is worthwhile to mention that $s_b [\lambda B^0 + \rho \dot{\beta}] = 0$. Hence, these terms (i.e., $-(\lambda B^0 + \rho \dot{\beta})$) remain intact and they are present in the expression for $Q_b^{(1)}$. We follow *exactly* the above *prescription* of our proposal in the case of the *non-nilpotent* Noether anti-BRST charge $Q_{ab}$ to obtain its *modified* off-shell nilpotent version as

$$\begin{aligned} Q_{ab} \longrightarrow Q_{ab}^{(1)} &= \int d^{D-1}x \left[\left(\partial^0 \bar{B}^i - \partial^i \bar{B}^0\right) \bar{C}_i + \left(\partial^0 C^i - \partial^i C^0\right) \partial_i \bar{\beta} \right. \\ &- \left. \left(\partial^0 \bar{C}^i - \partial^i \bar{C}^0\right) \bar{B}_i + \rho \bar{B}^0 - 2 \bar{\beta} \lambda - \lambda \dot{\bar{\beta}}\right], \end{aligned} \tag{63}$$

which satisfies the following relationship:

$$s_{ab} Q_{ab}^{(1)} = -i \left\{Q_{ab}^{(1)}, Q_{ab}^{(1)}\right\} = 0 \implies [Q_{ab}^{(1)}]^2 = 0. \tag{64}$$

The above observations establish the sanctity of our *proposal* that enables us to obtain *precisely* the off-shell nilpotent versions of the (anti-)BRST charges $[Q_{(a)b}^{(1)}]$ from the conserved Noether (anti-)BRST charges $[Q_{(a)b}]$ which are *non-nilpotent*.

We end this section with the following remarks. First, in the case of the 1D *massive* spinning particle, *only* the EL-EoMs with respect to the "gauge" and "supergauge" variables were good enough to convert the *non-nilpotent* Noether (anti-)BRST charges into the off-shell nilpotent versions of the (anti-)BRST conserved charges. Second, to obtain the off-shell nilpotent versions of the (anti-)BRST conserved charges in the context of the D-dimensional

non-Abelian one-form theory, we required Gauss's divergence theorem *plus* the EL-EoMs with respect to the gauge field. Third, in the context of the D-dimensional Abelian two-form theory, we invoked *twice* the Gauss divergence theorem and the EL-EoMs with respect to the gauge field and *fermionic* (anti-)ghost fields $(\bar{C}_\mu) C_\mu$ to obtain the off-shell nilpotent versions of the (anti-)BRST charges from the conserved Noether *non-nilpotent* (anti-)BRST charges. In all the above examples, we have *also* used the strength of the (anti-)BRST symmetry transformations, at appropriate places, so that we could obtain the off-shell *nilpotent* versions of the (anti-)BRST charges from the Noether conserved charges, which are *non-nilpotent*.

## 5. (Anti-)BRST Charges and Nilpotency: Arbitrary Dimensional St*ü*ckelberg-Modified Massive Abelian Three-Form Gauge Theory

We begin our present section with the (anti-)BRST invariant coupled (but equivalent) Lagrangian densities[7] for the St*ü*ckelberg-modified *massive* Abelian three-form gauge theory (see, e.g., [25] for details)

$$
\begin{aligned}
\mathcal{L}_B =\ & \mathcal{L}_S + (\partial_\mu A^{\mu\nu\lambda}) B_{\nu\lambda} - \frac{1}{2} B_{\mu\nu} B^{\mu\nu} + \frac{1}{2} B^{\mu\nu} \left[ \partial_\mu \phi_\nu - \partial_\nu \phi_\mu \mp m\, \Phi_{\mu\nu} \right] \\
& - (\partial_\mu \Phi^{\mu\nu}) B_\nu - \frac{1}{2} B^\mu B_\mu + \frac{1}{2} B^\mu \left[ \pm m\phi_\mu - \partial_\mu \phi \right] + \frac{m^2}{2} \bar{C}_{\mu\nu} C^{\mu\nu} \\
& + (\partial_\mu \bar{C}_{\nu\lambda} + \partial_\nu \bar{C}_{\lambda\mu} + \partial_\lambda \bar{C}_{\mu\nu})(\partial^\mu C^{\nu\lambda}) \pm m (\partial_\mu \bar{C}^{\mu\nu}) C_\nu \pm m \bar{C}^\nu (\partial^\mu C_{\mu\nu}) \\
& + (\partial_\mu \bar{C}_\nu - \partial_\nu \bar{C}_\mu) (\partial^\mu C^\nu) - \frac{1}{2} \left[ \pm m \bar{\beta}^\mu - \partial^\mu \bar{\beta} \right] \left[ \pm m \beta_\mu - \partial_\mu \beta \right] \\
& - (\partial_\mu \bar{\beta}_\nu - \partial_\nu \bar{\beta}_\mu) (\partial^\mu \beta^\nu) - \partial_\mu \bar{C}_2 \partial^\mu C_2 - m^2 \bar{C}_2 C_2 + [(\partial \cdot \bar{\beta}) \mp m \bar{\beta}] B \\
& - [(\partial \cdot \phi) \mp m \phi] B_1 - [(\partial \cdot \beta) \mp m \beta] B_2 + \left[ \partial_\nu \bar{C}^{\nu\mu} + \partial^\mu \bar{C}_1 \mp \frac{m}{2} \bar{C}^\mu \right] f_\mu \\
& - 2 F^\mu f_\mu - 2 F f - \left[ \partial_\nu C^{\nu\mu} + \partial^\mu C_1 \mp \frac{m}{2} C^\mu \right] F_\mu + \left[ \frac{1}{2} (\partial \cdot C) \mp m C_1 \right] F \\
& - \left[ \frac{1}{2} (\partial \cdot \bar{C}) \mp m \bar{C}_1 \right] f - B B_2 - \frac{1}{2} B_1^2,
\end{aligned}
\tag{65}
$$

$$
\begin{aligned}
\mathcal{L}_{\bar{B}} =\ & \mathcal{L}_S - (\partial_\mu A^{\mu\nu\lambda}) \bar{B}_{\nu\lambda} - \frac{1}{2} \bar{B}_{\mu\nu} \bar{B}^{\mu\nu} + \frac{1}{2} \bar{B}^{\mu\nu} \left[ \partial_\mu \phi_\nu - \partial_\mu \phi_\nu \pm m\, \Phi_{\mu\nu} \right] \\
& + (\partial_\mu \Phi^{\mu\nu}) \bar{B}_\nu - \frac{1}{2} \bar{B}^\mu \bar{B}_\mu + \frac{1}{2} \bar{B}^\mu \left[ \pm m\phi_\mu - \partial_\mu \phi \right] + \frac{m^2}{2} \bar{C}_{\mu\nu} C^{\mu\nu} \\
& + (\partial_\mu \bar{C}_{\nu\lambda} + \partial_\nu \bar{C}_{\lambda\mu} + \partial_\lambda \bar{C}_{\mu\nu})(\partial^\mu C^{\nu\lambda}) \pm m (\partial_\mu \bar{C}^{\mu\nu}) C_\nu \pm m \bar{C}^\nu (\partial^\mu C_{\mu\nu}) \\
& + (\partial_\mu \bar{C}_\nu - \partial_\nu \bar{C}_\mu) (\partial^\mu C^\nu) - \frac{1}{2} \left[ \pm m \bar{\beta}^\mu - \partial^\mu \bar{\beta} \right] \left[ \pm m \beta_\mu - \partial_\mu \beta \right] \\
& - (\partial_\mu \bar{\beta}_\nu - \partial_\nu \bar{\beta}_\mu) (\partial^\mu \beta^\nu) - \partial_\mu \bar{C}_2 \partial^\mu C_2 - m^2 \bar{C}_2 C_2 + [(\partial \cdot \bar{\beta}) \mp m \bar{\beta}] B \\
& - [(\partial \cdot \phi) \mp m \phi] B_1 - [(\partial \cdot \beta) \mp m \beta] B_2 + \left[ \partial_\nu C^{\nu\mu} - \partial^\mu C_1 \mp \frac{m}{2} C^\mu \right] \bar{f}_\mu \\
& + 2 \bar{F}^\mu \bar{f}_\mu + 2 \bar{F} \bar{f} - \left[ \partial_\nu \bar{C}^{\nu\mu} - \partial^\mu \bar{C}_1 \mp \frac{m}{2} \bar{C}^\mu \right] \bar{F}_\mu + \left[ \frac{1}{2} (\partial \cdot \bar{C}) \pm m \bar{C}_1 \right] \bar{F} \\
& - \left[ \frac{1}{2} (\partial \cdot C) \pm m C_1 \right] \bar{f} - B B_2 - \frac{1}{2} B_1^2,
\end{aligned}
\tag{66}
$$

where $\mathcal{L}_S$ is the St*ü*ckelberg-modified *classical* Lagrangian density that incorporates into it the totally antisymmetric tensor gauge field $(A_{\mu\nu\lambda})$ and antisymmetric $(\Phi_{\mu\nu} = -\Phi_{\nu\mu})$ St*ü*ckelberg field $(\Phi_{\mu\nu})$ along with *their* kinetic terms as follows [25]:

$$
\mathcal{L}_S = \frac{1}{24} H^{\mu\nu\lambda\zeta} H_{\mu\nu\lambda\zeta} - \frac{m^2}{6} A^{\mu\nu\lambda} A_{\mu\nu\lambda} \pm \frac{m}{3} A^{\mu\nu\lambda} \Sigma_{\mu\nu\lambda} - \frac{1}{6} \Sigma^{\mu\nu\lambda} \Sigma_{\mu\nu\lambda}.
\tag{67}
$$

In the above, the *kinetic* term with the tensor field $H_{\mu\nu\lambda\zeta}$ for the gauge field $A_{\mu\nu\lambda}$ owes its origin to the exterior derivative $d = d\, x^\mu\, \partial_\mu$ $(\mu = 0, 1 \ldots D - 1)$ because the Abelian four-form

is: $H^{(4)} = d A^{(3)} = \left[(d x^\mu \wedge d x^\nu \wedge d x^\lambda \wedge d x^\xi)/4!\right] H_{\mu\nu\lambda\xi}$ where the Abelian three-form *totally* antisymmetric gauge field is defined through: $A^{(3)} = \left[(d x^\mu \wedge d x^\nu \wedge d x^\lambda)/3!\right] A_{\mu\nu\lambda}$. In addition, the Abelian three-form $\Sigma^{(3)} = \left[(d x^\mu \wedge d x^\nu \wedge d x^\lambda)/3!\right] \Sigma_{\mu\nu\lambda}$ is defined through the Abelian Stückelberg two-form $\Phi^{(2)} = \left[(d x^\mu \wedge d x^\nu)/2!\right] \Phi_{\mu\nu}$ as: $\Sigma^{(3)} = d \Phi^{(2)}$, which implies that $\Sigma_{\mu\nu\lambda} = \partial_\mu \Phi_{\nu\lambda} + \partial_\nu \Phi_{\lambda\mu} + \partial_\lambda \Phi_{\mu\nu}$. It is *also* evident that $H_{\mu\nu\lambda\xi}$ is equal to

$$H_{\mu\nu\lambda\zeta} = \partial_\mu A_{\nu\lambda\zeta} - \partial_\nu A_{\lambda\zeta\mu} + \partial_\lambda A_{\zeta\mu\nu} - \partial_\zeta A_{\mu\nu\lambda}, \tag{68}$$

which is invoked in the definition of the kinetic term for the gauge field $A_{\mu\nu\lambda}$.

In the (anti-)BRST invariant Lagrangian densities $\mathcal{L}_B$ and $\mathcal{L}_{\bar{B}}$, we have the bosonic auxiliary fields $(B_{\mu\nu}, \bar{B}_{\mu\nu}, B_\mu, \bar{B}_\mu \, B_1, B_2, B)$ and a *fermionic* set of auxiliary fields is $(F_\mu, \bar{F}_\mu, f_\mu, \bar{f}_\mu, F, \bar{F}, f, \bar{f})$ out of which the two *bosonic* auxiliary fields $(B, B_2)$ carry the ghost numbers $(-2, +2)$, respectively, and a set of *fermionic* auxiliary fields $(F_\mu, \bar{f}_\mu, \bar{f}, F)$ carry the ghost number $(-1)$. On the other hand, a *fermionic* set of auxiliary fields $(\bar{F}_\mu, f_\mu, f, \bar{F})$ is endowed with ghost number $(+1)$. To maintain the *unitarity* in the theory, we need the *fermionic* set of ghost fields $(C_{\mu\nu}, \bar{C}_{\mu\nu}, C_\mu, \bar{C}_\mu, \bar{C}_2, C_2, \bar{C}_1, C_1)$ as well as the *bosonic* set of (anti-)ghost fields $(\bar{\beta}_\mu, \beta_\mu, \bar{\beta}, \beta)$ where the *latter* (anti-)ghost fields carry the ghost numbers $(-2, +2)$. To be precise, the set of bosonic anti-ghost fields $(\bar{\beta}_\mu, \bar{\beta})$ and the ghost fields $(\beta_\mu, \beta)$ are endowed with $(-2)$ and $(+2)$ ghost numbers, respectively. The fermionic set of (anti-)ghost fields $(\bar{C}_2) C_2$ has the ghost numbers $(-3) + 3$, respectively, and all the *rest* of the fermionic (anti-)ghost fields: $(\bar{C}_\mu, \bar{C}_1)$ and $(C_\mu, C_1)$ carry the ghost numbers $(-1)$ and $(+1)$, respectively. We have the bosonic vector and scalar fields $(\phi_\mu, \phi)$, too, in our theory, which appear in the gauge-fixing terms.

The following off-shell nilpotent $[s^2_{(a)b} = 0]$ anti-BRST transformations $[s_{(a)b}]$

$$s_{ab} A_{\mu\nu\lambda} = \partial_\mu \bar{C}_{\nu\lambda} + \partial_\nu \bar{C}_{\lambda\mu} + \partial_\lambda \bar{C}_{\mu\nu}, \quad s_{ab} \bar{C}_{\mu\nu} = \partial_\mu \bar{\beta}_\nu - \partial_\nu \bar{\beta}_\mu,$$
$$s_{ab} B_{\mu\nu} = -(\partial_\mu F_\nu - \partial_\nu F_\mu) \equiv (\partial_\mu \bar{f}_\nu - \partial_\nu \bar{f}_\mu), \quad s_{ab} C_{\mu\nu} = \bar{B}_{\mu\nu},$$
$$s_{ab} \Phi_{\mu\nu} = \pm m \, \bar{C}_{\mu\nu} - (\partial_\mu \bar{C}_\nu - \partial_\nu \bar{C}_\mu), \quad s_{ab} F_\mu = -\partial_\mu B_2,$$
$$s_{ab} \bar{C}_\mu = \pm m \, \bar{\beta}_\mu - \partial_\mu \bar{\beta}, \quad s_{ab} \phi_\mu = \bar{f}_\mu, \quad s_{ab} \bar{\beta}_\mu = \partial_\mu \bar{C}_2,$$

$$s_{ab} B_\mu = \pm m \, \bar{f}_\mu - \partial_\mu \bar{f}, \quad s_{ab} \beta_\mu = \bar{F}_\mu, \quad s_{ab} f_\mu = \partial_\mu B_1,$$
$$s_{ab} C_\mu = \bar{B}_\mu, \quad s_{ab} f = \pm m B_1, \quad s_{ab} \phi = \bar{f},$$
$$s_{ab} \bar{\beta} = \pm m \, \bar{C}_2, \quad s_{ab} \beta = \bar{F}, \quad s_{ab} F = \mp m B_2$$
$$s_{ab} \bar{C}_1 = -B_2, \quad s_{ab} C_1 = B_1, \quad s_{ab} C_2 = B,$$
$$s_{ab} \left[ H_{\mu\nu\lambda\zeta}, \bar{B}_{\mu\nu}, \bar{B}_\mu, \bar{F}_\mu, \bar{f}_\mu, \bar{F}, \bar{f}, B, B_1, B_2, \bar{C}_2 \right] = 0, \tag{69}$$

are the *symmetry* transformations for the action integral $S_1 = \int d^{D-1} x \, \mathcal{L}_{\bar{B}}$ because the Lagrangian density $\mathcal{L}_{\bar{B}}$ transforms to a *total* spacetime derivative under the infinitesimal, continuous and off-shell nilpotent $(s^2_{ab} = 0)$ anti-BRST symmetry transformations $s_{ab}$:

$$
\begin{aligned}
s_{ab} \mathcal{L}_{\bar{B}} = \;& \partial_\mu \Big[ \bar{B}^{\mu\nu} \bar{f}_\nu - (\partial^\mu \bar{C}^{\nu\lambda} + \partial^\nu \bar{C}^{\lambda\mu} + \partial^\lambda \bar{C}^{\mu\nu}) \bar{B}_{\nu\lambda} + B \, \partial^\mu \bar{C}_2 \\
& - B_2 \bar{F}^\mu - B_1 \bar{f}^\mu - (\partial^\mu \bar{\beta}^\nu - \partial^\nu \bar{\beta}^\mu) \bar{F}_\nu + \frac{1}{2} (\pm m \, \bar{\beta}^\mu - \partial^\mu \bar{\beta}) \bar{F} \\
& - (\partial^\mu \bar{C}^\nu - \partial^\nu \bar{C}^\mu) \bar{B}_\nu \mp m \, \bar{B}^{\mu\nu} \bar{C}_\nu - \frac{1}{2} \bar{B}^\mu \bar{f} \\
& \pm m \, C^{\mu\nu} (\pm m \, \bar{\beta}_\nu - \partial_\nu \bar{\beta}) \pm m \, (\partial^\mu \bar{\beta}^\nu - \partial^\nu \bar{\beta}^\mu) C_\nu \Big]. \tag{70}
\end{aligned}
$$

On the other hand, under the infinitesimal, continuous and off-shell nilpotent $(s_b^2 = 0)$ BRST *symmetry* transformations $(s_b)$

$$s_b A_{\mu\nu\lambda} = \partial_\mu C_{\nu\lambda} + \partial_\nu C_{\lambda\mu} + \partial_\lambda C_{\mu\nu}, \quad s_b C_{\mu\nu} = \partial_\mu \beta_\nu - \partial_\nu \beta_\mu,$$
$$s_b \bar{B}_{\mu\nu} = - (\partial_\mu \bar{F}_\nu - \partial_\nu \bar{F}_\mu) \equiv (\partial_\mu f_\nu - \partial_\nu f_\mu), \quad s_b \bar{C}_{\mu\nu} = B_{\mu\nu},$$
$$s_b \Phi_{\mu\nu} = \pm m\, C_{\mu\nu} - (\partial_\mu C_\nu - \partial_\nu C_\mu), \qquad s_b \bar{F}_\mu = -\partial_\mu B,$$
$$s_b C_\mu = \pm m\, \beta_\mu - \partial_\mu \beta, \qquad s_b \phi_\mu = f_\mu, \qquad s_b \beta_\mu = \partial_\mu C_2,$$
$$s_b \bar{B}_\mu = \pm m\, f_\mu - \partial_\mu f, \qquad s_b \bar{f}_\mu = -\partial_\mu B_1, \quad s_b \bar{\beta}_\mu = F_\mu,$$
$$s_b \bar{C}_\mu = B_\mu, \qquad s_b \bar{C}_2 = B_2, \qquad s_b \bar{C}_1 = -B_1,$$
$$s_b \phi = f, \qquad s_b \beta = \pm m\, C_2, \qquad s_b \bar{\beta} = F,$$
$$s_b \bar{F} = \mp m\, B, \qquad s_b \bar{f} = \mp m\, B_1, \qquad s_b C_1 = -B,$$
$$s_b [H_{\mu\nu\lambda\zeta}, B_{\mu\nu}, B_\mu, f_\mu, F_\mu, F, f, B, B_1, B_2, C_2] = 0, \tag{71}$$

the Lagrangian density $\mathcal{L}_B$ transforms to a *total* spacetime derivative

$$\begin{aligned}
s_b \mathcal{L}_B &= \partial_\mu \Big[ (\partial^\mu C^{\nu\lambda} + \partial^\nu C^{\lambda\mu} + \partial^\lambda C^{\mu\nu}) B_{\nu\lambda} + B^{\mu\nu} f_\nu - B_2\, \partial^\mu C_2 \\
&\quad - B_1\, f^\mu + B\, F^\mu - (\partial^\mu \beta^\nu - \partial^\nu \beta^\mu)\, F_\nu + \frac{1}{2}\, (\pm m\, \beta^\mu - \partial^\mu \beta)\, F \\
&\quad + (\partial^\mu C^\nu - \partial^\nu C^\mu)\, B_\nu \pm m\, B^{\mu\nu} C_\nu - \frac{1}{2}\, B^\mu f \\
&\quad \mp m\, \bar{C}^{\mu\nu} (\pm m\, \beta_\nu - \partial_\nu \beta) \mp m\, (\partial^\mu \beta^\nu - \partial^\nu \beta^\mu)\, \bar{C}_\nu \Big], 
\end{aligned} \tag{72}$$

which establishes the fact that the action integral $S_2 = \int d^{D-1} x\, \mathcal{L}_B$ respects the above infinitesimal, continuous and nilpotent BRST symmetry transformations $(s_b)$.

According to Noether's theorem, the above observations imply that the BRST *conserved* currents can be derived, by exploiting the standard theoretical formula, as [26]:

$$\begin{aligned}
J_{(b)}^\mu &= H^{\mu\nu\lambda\zeta} (\partial_\nu C_{\lambda\zeta}) \pm \frac{m}{2} [\pm m\, \bar{\beta}^\mu - \partial^\mu \bar{\beta}]\, C_2 \pm m\, \bar{C}^{\mu\nu} (\pm m\, \beta_\nu - \partial_\nu \beta) + (\partial^\mu C^{\nu\lambda} \\
&\quad + \partial^\nu C^{\lambda\mu} + \partial^\lambda C^{\mu\nu}) B_{\nu\lambda} - (\partial^\mu \bar{C}^{\nu\lambda} + \partial^\nu \bar{C}^{\lambda\mu} + \partial^\lambda \bar{C}^{\mu\nu}) (\partial_\nu \beta_\lambda - \partial_\lambda \beta_\nu) \mp m\, C^{\mu\nu} B_\nu \\
&\quad - B_1\, f^\mu + [\pm m\, A^{\mu\nu\lambda} - \Sigma^{\mu\nu\lambda}] [\pm m\, C_{\nu\lambda} - (\partial_\nu C_\lambda - \partial_\lambda C_\nu)] - B_2\, \partial^\mu C_2 + B^{\mu\nu} f_\nu \\
&\quad - (\partial^\mu \bar{\beta}^\nu - \partial^\nu \bar{\beta}^\mu) (\partial_\nu C_2) - (\partial^\mu \beta^\nu - \partial^\nu \beta^\mu)\, F_\nu + (\partial^\mu C^\nu - \partial^\nu C^\mu)\, B_\nu + B\, F^\mu \\
&\quad - \frac{1}{2}\, B^\mu f + \frac{1}{2}\, (\pm m\, \beta^\mu - \partial^\mu \beta)\, F - (\partial^\mu \bar{C}^\nu - \partial^\nu \bar{C}^\mu) (\pm m\, \beta_\nu - \partial_\nu \beta).
\end{aligned} \tag{73}$$

In exactly similar fashion, we obtain the *precise* expression for the conserved $(\partial_\mu J_{(ab)}^\mu = 0)$ anti-BRST Noether current $J_{(ab)}^\mu$ [26]:

$$\begin{aligned}
J_{(ab)}^\mu &= H^{\mu\nu\lambda\zeta} (\partial_\nu \bar{C}_{\lambda\zeta}) \pm \frac{m}{2} [\pm m\, \beta^\mu - \partial^\mu \beta]\, \bar{C}_2 \mp m\, C^{\mu\nu} (\pm m\, \bar{\beta}_\nu - \partial_\nu \bar{\beta}) - (\partial^\mu \bar{C}^{\nu\lambda} \\
&\quad + \partial^\nu \bar{C}^{\lambda\mu} + \partial^\lambda \bar{C}^{\mu\nu}) \bar{B}_{\nu\lambda} + (\partial^\mu C^{\nu\lambda} + \partial^\nu C^{\lambda\mu} + \partial^\lambda C^{\mu\nu}) (\partial_\nu \bar{\beta}_\lambda - \partial_\lambda \bar{\beta}_\nu) \pm m\, \bar{C}^{\mu\nu} \bar{B}_\nu \\
&\quad - \bar{f}^\mu B_1 + [\pm m\, A^{\mu\nu\lambda} - \Sigma^{\mu\nu\lambda}] [\pm m\, \bar{C}_{\nu\lambda} - (\partial_\nu \bar{C}_\lambda - \partial_\lambda \bar{C}_\nu)] + B\, \partial^\mu \bar{C}_2 + \bar{B}^{\mu\nu} \bar{f}_\nu \\
&\quad - (\partial^\mu \beta^\nu - \partial^\nu \beta^\mu) (\partial_\nu \bar{C}_2) - (\partial^\mu \bar{\beta}^\nu - \partial^\nu \bar{\beta}^\mu)\, \bar{F}_\nu - (\partial^\mu \bar{C}^\nu - \partial^\nu \bar{C}^\mu)\, \bar{B}_\nu - B_2\, \bar{F}^\mu \\
&\quad - \frac{1}{2}\, \bar{B}^\mu \bar{f} + \frac{1}{2}\, (\pm m\, \bar{\beta}^\mu - \partial^\mu \bar{\beta})\, \bar{F} + (\partial^\mu C^\nu - \partial^\nu C^\mu) (\pm m\, \bar{\beta}_\nu - \partial_\nu \bar{\beta}).
\end{aligned} \tag{74}$$

The conservation law $(\partial_\mu J_{(a)b}^\mu = 0)$ can be checked by using the EL-EoMs that have been derived in our earlier work [26]. In these proofs, the algebra is a bit involved, but it is quite straightforward to check that $\partial_\mu J_{(r)}^\mu = 0$ with $r = b, ab$.

The conserved Noether charges $Q_{(a)b} = \int d^{D-1}x \, J^0_{(a)b}$ (that emerge out from the conserved currents) $J^\mu_{(a)b}$ are as follows [26]:

$$
\begin{aligned}
Q_{ab} \; = \; & \int d^{D-1}x \, J^0_{(ab)} \equiv \int d^{D-1}x \, \Big[ H^{0ijk} \, (\partial_i \, \bar{C}_{jk}) \pm \frac{m}{2} \, [\pm m \, \beta^0 - \partial^0 \, \beta] \, \bar{C}_2 \\
& - \; [\pm m \, C^{0i} - (\partial^0 \, C^i - \partial^i \, C^0)] \, (\pm m \, \bar{\beta}_i - \partial_i \, \bar{\beta}) + [\pm m \, \bar{C}^{0i} - (\partial^0 \, \bar{C}^i - \partial^i \, \bar{C}^0)] \, \bar{B}_i \\
& - \; (\partial^0 \, \bar{C}^{ij} + \partial^i \, \bar{C}^{j0} + \partial^j \, \bar{C}^{0i}) \, \bar{B}_{ij} + (\partial^0 \, C^{ij} + \partial^i \, C^{j0} + \partial^j \, C^{0i}) \, (\partial_i \, \bar{\beta}_j - \partial_j \, \bar{\beta}_i) \\
& - \; \bar{f}^0 \, B_1 + [\pm m \, A^{0ij} - \Sigma^{0ij}] \, [\pm m \, \bar{C}_{ij} - (\partial_i \, \bar{C}_j - \partial_j \, \bar{C}_i)] + B \, \partial^0 \, \bar{C}_2 - \frac{1}{2} \, \bar{B}^0 \, \bar{f} - B_2 \, \bar{F}^0 \\
& + \; \bar{B}^{0i} \, \bar{f}_i - (\partial^0 \, \beta^i - \partial^i \, \beta^0) \, (\partial_i \, \bar{C}_2) - (\partial^0 \, \bar{\beta}^i - \partial^i \, \bar{\beta}^0) \, \bar{F}_i + \frac{1}{2} \, (\pm m \, \bar{\beta}^0 - \partial^0 \, \bar{\beta}) \, \bar{F} \Big],
\end{aligned}
\tag{75}
$$

$$
\begin{aligned}
Q_b \; = \; & \int d^{D-1}x \, J^0_{(b)} \equiv \int d^{D-1}x \, \Big[ H^{0ijk} \, (\partial_i \, C_{jk}) \pm \frac{m}{2} \, [\pm m \, \bar{\beta}^0 - \partial^0 \, \bar{\beta}] \, C_2 \\
& + \; [\pm m \, \bar{C}^{0i} - (\partial^0 \, \bar{C}^i - \partial^i \, \bar{C}^0)] \, (\pm m \, \beta_i - \partial_i \, \beta) - [\pm m \, C^{0i} - (\partial^0 \, C^i - \partial^i \, C^0)] \, B_i \\
& + \; (\partial^0 \, C^{ij} + \partial^i \, C^{j0} + \partial^j \, C^{0i}) \, B_{ij} - (\partial^0 \, \bar{C}^{ij} + \partial^i \, \bar{C}^{j0} + \partial^j \, \bar{C}^{0i}) \, (\partial_i \, \beta_j - \partial_j \, \beta_i) \\
& - \; B_1 \, f^0 + [\pm m \, A^{0ij} - \Sigma^{0ij}] \, [\pm m \, C_{ij} - (\partial_i \, C_j - \partial_j \, C_i)] - B_2 \, \partial^0 \, C_2 - \frac{1}{2} \, B^0 \, f + B^{0i} \, f_i \\
& - \; (\partial^0 \, \bar{\beta}^i - \partial^i \, \bar{\beta}^0) \, (\partial_i \, C_2) - (\partial^0 \, \beta^i - \partial^i \, \beta^0) \, F_i + B \, F^0 + \frac{1}{2} \, (\pm m \, \beta^0 - \partial^0 \, \beta) \, F \Big].
\end{aligned}
\tag{76}
$$

These conserved charges are the generators of *all* the (anti-)BRST symmetry transformations. However, they are *not* off-shell nilpotent of order two. Exploiting the principle behind the continuous symmetry transformations and their generators, we obtain the following

$$
\begin{aligned}
s_b \, Q_b = - i \, \{Q_b, \, Q_b\} \; = \; & \int d^{D-1}x \, \Big[ \pm \frac{m}{2} \, \big( \pm m \, F^0 - \partial^0 F \big) \, C_2 - (\partial^0 F^i - \partial^i F^0) \, (\partial_i C_2) \\
& - \; (\partial^0 \, B^{ij} + \partial^i \, B^{j0} + \partial^j \, B^{0i}) \, (\partial_i \beta_j - \partial_j \beta_i) \\
& + \; \big[ \pm m \, B^{0i} - (\partial^0 B^i - \partial^i B^0) \big] \, (\pm m \, \beta_i - \partial_i \beta) \Big] \neq 0, \\
s_{ab} \, Q_{ab} = - i \, \{Q_{ab}, \, Q_{ab}\} \; = \; & \int d^{D-1}x \, \Big[ \pm \frac{m}{2} \, \big( \pm m \, \bar{F}^0 - \partial^0 \bar{F} \big) \, \bar{C}_2 - (\partial^0 \bar{F}^i - \partial^i \bar{F}^0) \, (\partial_i \bar{C}_2) \\
& + \; (\partial^0 \, \bar{B}^{ij} + \partial^i \, \bar{B}^{j0} + \partial^j \, \bar{B}^{0i}) \, (\partial_i \bar{\beta}_j - \partial_j \bar{\beta}_i) \\
& - \; \big[ \pm m \, \bar{B}^{0i} - (\partial^0 \bar{B}^i - \partial^i \bar{B}^0) \big] \, (\pm m \, \bar{\beta}_i - \partial_i \bar{\beta}) \Big] \neq 0,
\end{aligned}
\tag{77}
$$

where the l.h.s. of the above equations has been explicitly computed by using the (anti-)BRST symmetry transformations (cf. Equations (69) and (71)) and the expressions for the Noether conserved charges $Q_{(a)b}$ (cf. Equations (75) and (76)). The above observations establish that the conserved Noether (anti-)BRST charges $Q_{(a)b}$ are *not* off-shell nilpotent of order two.

We now follow our step-by-step *proposal* to obtain the off-shell nilpotent expressions for the (anti-)BRST charges $Q^{(1)}_{(a)b}$ where the off-shell nilpotency is proven by using the principle behind the continuous symmetry transformations and their generators as: $s_b \, Q^{(1)}_b = - i \, \{Q^{(1)}_b, \, Q^{(1)}_b\} = 0$ and $s_{ab} \, Q^{(1)}_{ab} = - i \, \{Q^{(1)}_{ab}, \, Q^{(1)}_{ab}\} = 0$. First of all, we focus on the BRST charge $Q_b$. The *first* step is to use the equation of motion with respect to the gauge field. Toward this goal in mind, we note that the *first* term of $Q_b$ can be re-expressed as:

$$
\int d^{D-1}x \, \Big[ H^{0ijk} \, \partial_i \, C_{jk} \Big] = \int d^{D-1}x \, \Big[ \partial_i \, \{H^{0ijk} \, C_{jk}\} \Big] - \int d^{D-1}x \, \Big[ (\partial_i \, H^{0ijk}) \, C_{jk} \Big].
\tag{78}
$$

The *first* term on the r.h.s. of the above equation *vanishes* for the physical fields due to celebrated Gauss's divergence theorem. In the *second* term, we apply the following equation of motion with respect to the gauge field $A_{\mu\nu\lambda}$, namely:

$$\partial_\rho H^{\rho\mu\nu\lambda} + m^2 A^{\mu\nu\lambda} \mp m \Sigma^{\mu\nu\lambda} + (\partial^\mu B^{\nu\lambda} + \partial^\nu B^{\lambda\mu} + \partial^\lambda B^{\mu\nu}) = 0$$

$$\implies -(\partial_i H^{0ijk}) C_{jk} = \mp m \left[ \pm m A^{0ij} - \Sigma^{0ij} \right] C_{ij} - (\partial^0 B^{ij} + \partial^i B^{j0} + \partial^j B^{oi}) C_{ij}. \tag{79}$$

In the *second* step, we look for the terms of $Q_b$ that cancel out with some of the terms that emerge out after the *first* step. In this context, we note that such an appropriate term is

$$+ \left[ \pm m A^{0ij} - \Sigma^{0ij} \right] \left[ \pm m C_{ij} - (\partial_i C_j - \partial_j C_i) \right]. \tag{80}$$

The sum of (79) and (80) yields the following:

$$\begin{aligned} &- (\partial_i H^{0ijk}) C_{jk} + \left[ \pm m A^{0ij} - \Sigma^{0ij} \right] \left[ \pm m C_{ij} - (\partial_i C_j - \partial_j C_i) \right] \\ &\equiv -(\partial^0 B^{ij} + \partial^i B^{j0} + \partial^j B^{oi}) C_{ij} - 2 \left[ \pm m A^{0ij} - \Sigma^{0ij} \right] (\partial_i C_j). \end{aligned} \tag{81}$$

We comment here that as pointed out earlier, one of the key ingredients of our proposal is the use of EL-EoM with respect to the gauge field. This step opens up a *decisive* door for (i) our further applications of Gauss's divergence theorem, (ii) use of the appropriate EL-EoM, and (iii) the application of the BRST symmetry transformations ($s_b$) at appropriate places. In the *third* step, we focus on the *second* term of the r.h.s., which can be written as:

$$-2 \partial_i \left[ (\pm m A^{0ij} - \Sigma^{0ij}) C_j \right] + 2 \partial_i \left[ (\pm m A^{0ij} - \Sigma^{0ij}) \right] C_j. \tag{82}$$

The terms in (82) are inside the integration. Thus, the *first* term of (82) will *vanish* due to Gauss's divergence theorem for *physical* fields. Using the following equation of motion with respect to the Stückelberg field $\Phi_{\mu\nu}$ from the Lagrangian density $\mathcal{L}_B$, we obtain:

$$\partial_\mu \Sigma^{\mu\nu\lambda} \mp m (\partial_\mu A^{\mu\nu\lambda}) + \frac{1}{2} (\partial^\nu B^\lambda - \partial^\lambda B^\nu) \mp \frac{m}{2} B^{\nu\lambda} = 0$$

$$\implies 2 \partial_i \left[ \pm m A^{0ij} - \Sigma^{0ij} \right] C_j = \pm m B^{0i} C_i - (\partial^0 B^i - \partial^i B^0) C_i, \tag{83}$$

where the l.h.s. is nothing but the *second* term of (82). Using (83), we can re-write the whole *sum* of (81) as follows:

$$+ \left[ \pm m B^{0i} - (\partial^0 B^i - \partial^i B^0) \right] C_i - (\partial^0 B^{ij} + \partial^i B^{j0} + \partial^j B^{oi}) C_{ij}. \tag{84}$$

Within the framework of our proposal, the terms in (84) will be present in the off-shell nilpotent form of the BRST charge ($Q_b^{(1)}$) because they have come out from the use of EL-EoMs and, hence, will remain *intact*. This is also required due to the fact that, as pointed out earlier, we wish to obtain $s_b Q_b^{(1)} = 0$ which will automatically imply that $[Q_b^{(1)}]^2 = 0$. In the *fourth* step, we apply the BRST symmetry transformation ($s_b$) on (84), which leads to the following explicit expression:

$$\begin{aligned} &+ \left[ \pm m B^{0i} - (\partial^0 B^i - \partial^i B^0) \right] (\pm m \beta_i - \partial_i \beta) \\ &- (\partial^0 B^{ij} + \partial^i B^{j0} + \partial^j B^{oi}) (\partial_i \beta_j - \partial_j \beta_i). \end{aligned} \tag{85}$$

In the *fifth* step, we modify some of the terms of $Q_b$ so that *when $s_b$ acts on a part of them*, there is precise cancellation between the ensuing result and (85). In this context, we modify the following terms from the explicit expression for $Q_b$ (cf. Equation (76)), namely:

$$- \quad (\partial^0 \bar{C}^{ij} + \partial^i \bar{C}^{j0} + \partial^j \bar{C}^{oi})(\partial_i \beta_j - \partial_j \beta_i) = +(\partial^0 \bar{C}^{ij} + \partial^i \bar{C}^{j0} + \partial^j \bar{C}^{oi})(\partial_i \beta_j - \partial_j \beta_i)$$
$$- \quad 2(\partial^0 \bar{C}^{ij} + \partial^i \bar{C}^{j0} + \partial^j \bar{C}^{oi})(\partial_i \beta_j - \partial_j \beta_i),$$
$$+ \quad \left[ \pm m \bar{C}^{0i} - (\partial^0 \bar{C}^i - \partial^i \bar{C}^0) \right] (\pm m \beta_i - \partial_i \beta) = 2\left[ \pm m \bar{C}^{0i} - (\partial^0 \bar{C}^i \right.$$
$$- \quad \left. \partial^i \bar{C}^0) \right] (\pm m \beta_i - \partial_i \beta) - \left[ \pm m \bar{C}^{0i} - (\partial^0 \bar{C}^i - \partial^i \bar{C}^0) \right] (\pm m \beta_i - \partial_i \beta). \tag{86}$$

In the *sixth* step, we note that in the above modified expressions, when we apply $s_b$ on a part of (86) as explicitly written below, namely:

$$s_b \left[ (\partial^0 \bar{C}^{ij} + \partial^i \bar{C}^{j0} + \partial^j \bar{C}^{0i})(\partial_i \beta_j - \partial_j \beta_i) \right.$$
$$\left. - \{ \pm m \bar{C}^{0i} - (\partial^0 \bar{C}^i - \partial^i \bar{C}^0) \}(\pm m \beta_i - \partial_i \beta) \right], \tag{87}$$

we obtain the desired result

$$(\partial^0 B^{ij} + \partial^i B^{j0} + \partial^j B^{0i})(\partial_i \beta_j - \partial_j \beta_i)$$
$$- \{ \pm m B^{0i} - (\partial^0 B^i - \partial^i B^0) \}(\pm m \beta_i - \partial_i \beta) \right], \tag{88}$$

which *precisely* cancels out with whatever we have obtained in (85). At this juncture, we point out that in the off-shell nilpotent version of the BRST charge $Q_b^{(1)}$, the terms in the square bracket of (87) will be *always* (along with (84) (as pointed out earlier)) present. We now focus on the leftover terms of (86), which can be re-expressed as follows:

$$\pm 2m \{ \pm m \bar{C}^{0i} - (\partial^0 \bar{C}^i - \partial^i \bar{C}^0) \} \beta_i$$
$$- 2 \{ \pm m \bar{C}^{0i} - (\partial^0 \bar{C}^i - \partial^i \bar{C}^0) \} \partial_i \beta$$
$$- 4 \left[ \partial^0 \bar{C}^{ij} + \partial^i \bar{C}^{j0} + \partial^j \bar{C}^{0i} \right] (\partial_i \beta_j). \tag{89}$$

All the above terms are inside the integration. Hence, we can apply the following algebraic tricks on the *last* two terms of (89) to obtain the following

$$-2 \{ \pm m \bar{C}^{0i} - (\partial^0 \bar{C}^i - \partial^i \bar{C}^0) \} \partial_i \beta \quad = \quad \partial_i \left[ -2 \{ \pm m \bar{C}^{0i} - (\partial^0 \bar{C}^i - \partial^i \bar{C}^0) \} \beta \right]$$
$$+ \quad 2\partial_i \left[ \pm m \bar{C}^{0i} - (\partial^0 \bar{C}^i - \partial^i \bar{C}^0) \right] \beta$$
$$-4 \left[ \partial^0 \bar{C}^{ij} + \partial^i \bar{C}^{j0} + \partial^j \bar{C}^{0i} \right] \partial_i \beta_j \quad = \quad \partial_i \left[ -4 \{ \partial^0 \bar{C}^{ij} + \partial^i \bar{C}^{j0} + \partial^j \bar{C}^{0i} \} \beta_j \right]$$
$$+ \quad 4\partial_i \left[ \partial^0 \bar{C}^{ij} + \partial^i \bar{C}^{j0} + \partial^j \bar{C}^{0i} \right] \beta_j, \tag{90}$$

so that the *total* space derivative terms *vanish* due to Gauss's divergence theorem for the *physical* fields, and we are left with the following from the above Equation (90):

$$2\partial_i \left[ \pm m \bar{C}^{0i} - (\partial^0 \bar{C}^i - \partial^i \bar{C}^0) \right] \beta + 4\partial_i \left[ \partial^0 \bar{C}^{ij} + \partial^i \bar{C}^{j0} + \partial^j \bar{C}^{0i} \right] \beta_j. \tag{91}$$

In the above, we apply the following equation of motion:

$$\partial_\mu [\partial^\mu \bar{C}^\nu - \partial^\nu \bar{C}^\mu] - \frac{1}{2} \partial^\nu F \mp m (\partial_\mu \bar{C}^{\mu\nu}) \pm \frac{m}{2} F^\nu = 0,$$
$$\implies 2\partial_i [\pm m \bar{C}^{0i} - (\partial^0 \bar{C}^i - \partial^i \bar{C}^0)] \beta = -(\pm m F^0 - \partial^0 F) \beta, \tag{92}$$

which provides the *concise* value of the *first* term of (91). Exactly, in a similar fashion, the application of the following equation of motion

$$\partial_\mu \left[\partial^\mu \bar{C}^{\nu\lambda} + \partial^\nu \bar{C}^{\lambda\mu} + \partial^\lambda \bar{C}^{\mu\nu}\right] \pm \frac{m}{2}\left(\partial^\nu \bar{C}^\lambda - \partial^\lambda \bar{C}^\nu\right)$$

$$+ \frac{1}{2}\left(\partial^\nu F^\lambda - \partial^\lambda F^\nu\right) - \frac{m^2}{2}\bar{C}^{\nu\lambda} = 0,$$

$$\implies 4\,\partial_i \left[\partial^0 \bar{C}^{ij} + \partial^i \bar{C}^{j0} + \partial^j \bar{C}^{0i}\right]\beta_j = 2\left(\partial^0 F^i - \partial^i F^0\right)\beta_i$$

$$\mp 2\,m\left[\pm m\,\bar{C}^{0i} - \left(\partial^0 \bar{C}^i - \partial^i \bar{C}^0\right)\right]\beta_i, \tag{93}$$

leads to the alternative (but appropriate) form of the *second* term of (91). The *sum* of the r.h.s. of (92) and (93) and the leftover term (i.e., the *first* term) of (89) is equal to:

$$2\left(\partial^0 F^i - \partial^i F^0\right)\beta_i - \left(\pm m\,F^0 - \partial^0 F\right)\beta. \tag{94}$$

At this stage, we take the *seventh* step and apply the BRST symmetry transformations $(s_b)$ on (94) which yields the following explicit expression, namely:

$$- 2\left(\partial^0 F^i - \partial^i F^0\right)\left(\partial_i C_2\right) + \left(\pm m\,F^0 - \partial^0 F\right)\left(\pm m\,C_2\right). \tag{95}$$

We emphasize that the term in (94) will be present in the off-shell nilpotent version of the BRST charge $(Q_b^{(1)})$. We take now the *eighth* step and modify some of the terms of $Q_b$ so that when we apply the BRST transformations $(s_b)$ on a *part* of them, the resulting expressions *must* cancel out (95) in a *precise* manner. Toward this goal in mind, we *modify* the following appropriate terms of the *original* expression for $Q_b$ (cf. Equation (76)):

$$-\quad \left(\partial^0 \bar{\beta}^i - \partial^i \bar{\beta}^0\right)\partial_i C_2 \pm \frac{1}{2}\,m\left(\pm m\,\bar{\beta}^0 - \partial^0 \bar{\beta}\right)C_2$$

$$\equiv\quad 2\left(\partial^0 \bar{\beta}^i - \partial^i \bar{\beta}^0\right)\partial_i C_2 \mp m\left(\pm m\,\bar{\beta}^0 - \partial^0 \bar{\beta}\right)C_2$$

$$\pm\quad \frac{3}{2}\left(\pm m\,\bar{\beta}^0 - \partial^0 \bar{\beta}\right)C_2 - 3\left(\partial^0 \bar{\beta}^i - \partial^i \bar{\beta}^0\right)\partial_i C_2. \tag{96}$$

It is evident that if we apply the BRST symmetry transformations $(s_b)$ on the first *two* terms of (96), the resulting expressions will cancel out the terms that are written in (95). Hence, along with (94), the above *first* two terms will be present in the off-shell nilpotent version of the BRST charge $(Q_b^{(1)})$ so that our central objective $s_b\,Q_b^{(1)} = 0$ can be fulfilled. At this juncture, we focus on the *last* term of (96), which can be re-written as

$$-3\left(\partial^0 \bar{\beta}^i - \partial^i \bar{\beta}^0\right)\partial_i C_2 = \partial_i\left[-3\left(\partial^0 \bar{\beta}^i - \partial^i \bar{\beta}^0\right)C_2\right] + 3\,\partial_i\left[\left(\partial^0 \bar{\beta}^i - \partial^i \bar{\beta}^0\right)\right]C_2. \tag{97}$$

The above terms are inside the integral. Hence, the *first* term on the r.h.s. will *vanish* due to Gauss's divergence theorem and *only* the *second* term on the r.h.s. of (97) will survive. Now, we apply the following EL-EoM

$$\Box\,\bar{\beta}_\mu - \partial_\mu\left(\partial \cdot \bar{\beta}\right) + \partial_\mu B_2 - \frac{m^2}{2}\,\bar{\beta}_\mu \pm \frac{m}{2}\,\partial_\mu \bar{\beta} = 0$$

$$\implies\quad 3\,\partial_i\left[\partial^0 \bar{\beta}^i - \partial^i \bar{\beta}^0\right]C_2 = 3\,\dot{B}_2\,C_2 \mp \frac{3}{2}\,m\left(\pm m\,\bar{\beta}^0 - \partial^0 \bar{\beta}\right)C_2, \tag{98}$$

which will replace the last term of (96). The substitution of (98) into (96) yields the following concise and beautiful result, namely:

$$\pm \frac{3}{2}\,m\left(\pm m\,\bar{\beta}^0 - \partial^0 \bar{\beta}\right)C_2 + 3\,\partial_i\left[\partial^0 \bar{\beta}^i - \partial^i \bar{\beta}^0\right]C_2 = 3\,\dot{B}_2\,C_2. \tag{99}$$

If we apply *further* the BRST symmetry transformation $(s_b)$ on (99), it turns out to be *zero*. Here, *all* our steps *terminate* (according to our proposal). It is self-evident that the

r.h.s. of (99) will be part of $Q_b^{(1)}$. Ultimately, the off-shell *nilpotent* version $Q_b^{(1)}$, from the *non-nilpotent* Noether conserved charge $Q_b$, is as follows

$$
\begin{aligned}
Q_b \longrightarrow Q_b^{(1)} = \int d^{D-1}x \Big[ & \left(\partial^0 C^{ij} + \partial^i C^{j0} + \partial^j C^{0i}\right) B_{ij} - \left[\pm m\, C^{0i} - \left(\partial^0 C^i - \partial^i C^0\right)\right] B_i \\
& - \left(\partial^0 B^{ij} + \partial^i B^{j0} + \partial^j B^{0i}\right) C_{ij} + \left[\pm m\, B^{0i} - \left(\partial^0 B^i - \partial^i B^0\right)\right] C_i \\
& - \left[\pm m\, \bar{C}^{0i} - \left(\partial^0 \bar{C}^i - \partial^i \bar{C}^0\right) \left(\pm m\, \beta_i - \partial_i\beta\right)\right] + 2\left(\partial^0 \bar{\beta}^i - \partial^i \bar{\beta}^0\right) \partial_i C_2 \\
& \mp m\left(\pm m\, \bar{\beta}^0 - \partial^0 \bar{\beta}\right) C_2 + \left(\partial^0 \bar{C}^{ij} + \partial^i \bar{C}^{j0} + \partial^j \bar{C}^{0i}\right)\left(\partial_i\beta_j - \partial_j\beta_i\right) \\
& + 2\left(\partial^0 F^i - \partial^i F^0\right)\beta_i - \left(\pm m\, F^0 - \partial^0 F\right)\beta + 3\dot{B}_2\, C_2 - B_2\, \dot{C}_2 + B\, F^0 \\
& - B_1\, f^0 - \frac{1}{2} B^0 f + B^{0i} f_i + \frac{1}{2}\left(\pm m\, \beta^0 - \partial^0\beta\right) F - \left(\partial^0\beta^i - \partial^i\beta^0\right) F_i \Big].
\end{aligned}
\tag{100}
$$

It will be noted that we have *not* touched several terms of the Noether conserved charge $Q_b$ (cf. Equation (76)) because these terms are BRST invariant. For instance, we have the following explicit observations:

$$
s_b\left[\left(\partial^0 C^{ij} + \partial^i C^{j0} + \partial^j C^{0i}\right) B_{ij}\right] = 0, \quad s_b\left[\left\{\pm m\, C^{0i} - \left(\partial^0 C^i - \partial^i C^0\right)\right\} B_i\right] = 0,
$$
$$
s_b\left[\left(\pm m\, F^0 - \partial^0 F\right)\beta\right] = 0, \quad s_b\left(B_1\, f^0\right) = 0, \quad s_b\left(B^0 f\right) = 0, \quad s_b\left(B^{0i} f_i\right) = 0,
$$
$$
s_b\left[\left(\pm m\, \beta^0 - \partial^0\beta\right) F\right] = 0, \quad s_b\left[\left(\partial^0\beta^i - \partial^i\beta^0\right) F_i\right] = 0, \quad s_b\left(\dot{B}_2\, C_2\right) = 0,
$$
$$
s_b\left(B_2\, \dot{C}_2\right) = 0, \quad s_b\left(B\, F^0\right) = 0.
\tag{101}
$$

It is straightforward to check that the following is *true*, namely:

$$
s_b\, Q_b^{(1)} = -i\left\{Q_b^{(1)}, Q_b^{(1)}\right\} = 0 \implies \left[Q_b^{(1)}\right]^2 = 0,
\tag{102}
$$

where the l.h.s. is computed *directly* by using the BRST symmetry transformations (71) on the expression for the *modified* version of the BRST charge $Q_b^{(1)}$.

We follow the prescriptions and *proposal* outlined above to compute the exact expression for the off-shell *nilpotent* version of the anti-BRST charge $Q_{ab}^{(1)}$ from the *non-nilpotent* conserved Noether anti-BRST charge $Q_{ab}$ as follows:

$$
\begin{aligned}
Q_{ab} \longrightarrow Q_{ab}^{(1)} = \int d^{D-1}x \Big[ & \left(\partial^0 \bar{B}^{ij} + \partial^i \bar{B}^{j0} + \partial^j \bar{B}^{0i}\right) \bar{C}_{ij} - \left[\pm m\, \bar{B}^{0i} - \left(\partial^0 \bar{B}^i - \partial^i \bar{B}^0\right)\right] \bar{C}_i \\
& - \left(\partial^0 \bar{C}^{ij} + \partial^i \bar{C}^{j0} + \partial^j \bar{C}^{0i}\right) \bar{B}_{ij} + \left[\pm m\, \bar{C}^{0i} - \left(\partial^0 \bar{C}^i - \partial^i \bar{C}^0\right)\right] \bar{B}_i \\
& + \left[\pm m\, C^{0i} - \left(\partial^0 C^i - \partial^i C^0\right)\right]\left(\pm m\, \bar{\beta}_i - \partial_i\bar{\beta}\right) + 2\left(\partial^0 \beta^i - \partial^i \beta^0\right) \partial_i \bar{C}_2 \\
& \mp m\left(\pm m\, \beta^0 - \partial^0\beta\right) \bar{C}_2 - \left(\partial^0 C^{ij} + \partial^i C^{j0} + \partial^j C^{0i}\right)\left(\partial_i\bar{\beta}_j - \partial_j\bar{\beta}_i\right) \\
& + 2\left(\partial^0 \bar{F}^i - \partial^i \bar{F}^0\right)\bar{\beta}_i - \left(\pm m\, \bar{F}^0 - \partial^0 \bar{F}\right)\bar{\beta} - 3\dot{B}\, \bar{C}_2 + B\, \dot{\bar{C}}_2 - B_2\, \bar{F}^0 \\
& - B_1\, \bar{f}^0 - \frac{1}{2}\bar{B}^0 \bar{f} + \bar{B}^{0i} \bar{f}_i + \frac{1}{2}\left(\pm m\, \bar{\beta}^0 - \partial^0\bar{\beta}\right) \bar{F} - \left(\partial^0\bar{\beta}^i - \partial^i\bar{\beta}^0\right)\bar{F}_i \Big].
\end{aligned}
\tag{103}
$$

It is now straightforward to check that:

$$
s_{ab}\, Q_{ab}^{(1)} = -i\left\{Q_{ab}^{(1)}, Q_{ab}^{(1)}\right\} = 0 \implies \left[Q_{ab}^{(1)}\right]^2 = 0.
\tag{104}
$$

The above observation proves that we have derived the off-shell *nilpotent* version $\left[Q_{ab}^{(1)}\right]$ of the *non-nilpotent* Noether conserved charge $Q_{ab}$ in a precise and logical manner. Our final results are the expressions for $Q_{(a)b}^{(1)}$ in (104) and (100).

## 6. Conclusions

In our present investigation, for a few physically interesting *gauge* systems, we have shown that wherever there is existence of the coupled (but equivalent) Lagrangians/Lagrangian densities due to the presence of the (anti-)BRST invariant CF-type restriction(s), we observe that the Noether theorem does *not* lead to the derivation of the off-shell nilpotent versions of the conserved (anti-)BRST charges $[Q_{(a)b}]$ within the framework of BRST formalism. These Noether conserved charges are found to be the generators for the infinitesimal, continuous and off-shell nilpotent (anti-)BRST symmetry transformations from which they are derived by exploiting the theoretical strength of Noether's theorem. However, they are found to be non-nilpotent in the sense that $s_b Q_b = -i\{Q_b, Q_b\} \neq 0$ and $s_{ab} Q_{ab} = -i\{Q_{ab}, Q_{ab}\} \neq 0$, which can be verified by applying the (anti-)BRST transformations directly on the Noether conserved charges $Q_{(a)b}$. In other words, by *directly* computing the explicit expressions for $s_b Q_b$ and $s_{ab} Q_{ab}$, which are the l.h.s. of: $s_b Q_b = -i\{Q_b, Q_b\}$ and $s_{ab} Q_{ab} = -i\{Q_{ab}, Q_{ab}\}$, we show that these charges are *not* nilpotent (see, e.g., Equation (77)). However, a close and careful look at them demonstrates that $s_b Q_b$ and $s_{ab} Q_{ab}$ are proportional to the EL-EoM. For instance, we are sure that in the simple case of a BRST-invariant 1D massive spinning relativistic particle, the r.h.s. of (14) is zero due to the EL-EoMs (15) and (23) that are derived from $L_b$ and $L_{\bar{b}}$, respectively.

In the derivations of the off-shell nilpotent versions of the (anti-)BRST charges $[Q^{(1)}_{(a)b}]$, the crucial roles are played by (i) the Gauss divergence theorem, (ii) the appropriate EL-EoMs from the appropriate Lagrangian/Lagrangian density of a set of coupled (but equivalent) Lagrangians/Lagrangian densities, and (iii) the application of the (anti-)BRST symmetry transformations at appropriate places (cf. Section 5 for details). We lay emphasis on the fact that for the D-dimensional (D ≥ 2) higher $p$-form ($p = 1, 2, 3, \dots$) gauge theories, it is imperative to *first* exploit the Gauss divergence theorem so that we can use the EL-EoMs with respect to the *gauge* field. For the 1D case of a spinning (i.e., SUSY) relativistic particle, we have shown that the Gauss divergence theorem, for obvious reasons, is *not* required *at all* and we *directly* use the EL-EoMs, right in the beginning, with respect to the "gauge" and "supergauge" variables. This is *not* the case with the *rest* of the examples considered in our present investigation which are connected with the D-dimensional (D ≥ 2) (non-)Abelian $p$-form ($p = 1, 2, 3, \dots$) *massless* and *massive* gauge theories.

The sequence of our proposal, to obtain the off-shell *nilpotent* versions of the (anti-)BRST charges from the *non-nilpotent* Noether (anti-)BRST charges, is as follows. In the *first* step, we apply the Gauss divergence theorem and take the help of EL-EoM with respect to the gauge field. This is the *crucial* and key first step of our proposal. In the next (i.e., second) step, we observe carefully whether there are any addition, subtraction and/or cancellation of the resulting terms (from the *first* step) with any of the terms of the non-nilpotent Noether conserved charges. The existing terms, after these *two* steps, will *always* be present in the off-shell nilpotent version of the (anti-)BRST charges $Q^{(1)}_{(a)b}$. After the above two steps, we apply the (anti-)BRST symmetry transformations on the existing terms. In the *third* step, we modify some of the appropriate terms of the non-nilpotent Noether conserved (anti-) BRST charges and see to it that a part of these modified terms cancel *precisely* with the terms that have appeared after the application of the nilpotent (anti-)BRST transformations on the *existing* terms (after the *first two* steps of our proposal). The *parts* which participate in the above *cancellation* are *also* always present in the off-shell nilpotent versions of (anti-)BRST charges $[Q^{(1)}_{(a)b}]$. After this step, it is the *interplay* amongst the Gauss divergence theorem, appropriate[8] EL-EoMs and application of the nilpotent (anti-) BRST symmetry transformations at appropriate places that lead to the derivation of the *precise* forms of the off-shell nilpotent versions[9] of the (anti-)BRST charges $[Q^{(1)}_{(a)b}]$ from the *non-nilpotent* Noether conserved (anti-)BRST charges (cf. Sections 4 and 5 for details).

Before we end this section by pointing out our future directions of investigation in the next paragraph, it is worthwhile to point out the physical significance of the conserved and off-shell nilpotent (anti-)BRST charges in the context of a given gauge theory which

is endowed with a set of non-trivial CF-type restrictions(s). The physicality condition $(Q_B^{(1)} \mid phys > = 0$, cf. Section 3) ensures that the operator form of the first-class constraints of the given gauge theory annihilate the physical state, which is consistent with the requirements of the Dirac quantization condition for a theory endowed with constraints (see, e.g., [5,6] for details). As has been shown by Weinberg [19], in the Fock space, the gauge-transformed states differ from their original counterparts by the BRST-*exact* states. Hence, if we have an off-shell nilpotent BRST charge, their difference becomes *trivial* when we invoke the physicality condition on the states with respect to the BRST charge. This argument has been extended in our earlier research works (see, e.g., [28,29]) where we have been able to prove that the 2D (non-)Abelian one-form and 4D Abelian two-form gauge theories are the tractable field–theoretic models for the Hodge theory. In the case of the 2D (non-)Abelian one-form theories, we have been able to show that the BRST and co-BRST symmetry transformations "gauge away" both the d.o.f. of the gauge fields and these theories become a *new* kind of topological field theory (see, e.g., [30] for details).

We end this section with the final comment that our proposal is very *general* and it can be applied to any physical system where (i) the (anti-)BRST invariant non-trivial CF-type restrictions exist, and (ii) the coupled (but equivalent) Lagrangians/Lagrangian densities describe the dynamics of the above physical systems within the framework of BRST formalism. The systematic application of our proposal sheds light on the appropriate directions that should be followed in order to obtain the off-shell nilpotent versions of the (anti-)BRST charges from the non-nilpotent versions of the Noether conserved (anti-)BRST charges. We plan to extend our present ideas in the context of more challenging problems of physical interest in the future.

**Funding:** This research work received no external funding.

**Institutional Review Board Statement:** Not applicable.

**Informed Consent Statement:** Not applicable.

**Data Availability Statement:** No data were used to support this study.

**Acknowledgments:** One of us (AKR) thankfully acknowledges the financial support from the Institution of Eminence (IoE) *Research Incentive Grant* of PFMS Scheme No. 3254-World Class Institutions to which Banaras Hindu University, Varanasi, belongs. The present investigation has been carried out under the above *Research Incentive Grant* which has been launched by the Government of India. All the authors express their deep sense of gratitude to all the reviewers for their fruitful comments which have made our presentation more accurate and beautiful.

**Conflicts of Interest:** The authors declare no conflict of interest.

## Appendix A. On the Massive Abelian Two-Form Theory

To supplement our discussions in Section 4, in this Appendix A, we concisely mention the off-shell nilpotency property of the *non-nilpotent* Noether conserved (anti-)BRST charges for the Stückelberg-modified massive Abelian two-form gauge theory within the framework of BRST formalism. For this purpose, we begin with the generalized (i.e., Stückelberg-modified) versions of the (anti-)BRST invariant Lagrangian densities of Equations (46) and (47) for the D-dimensional *massive* Abelian two-form theory (see, e.g., [27,31] for details) as

$$\mathcal{L}_b = \frac{1}{12} H_{\mu\nu\eta} H^{\mu\nu\eta} - \frac{1}{4} m^2 B_{\mu\nu} B^{\mu\nu} - \frac{1}{4} \Phi_{\mu\nu} \Phi^{\mu\nu} + \frac{1}{2} m B_{\mu\nu} \Phi^{\mu\nu} - B^2$$

$$- B(\partial \cdot \phi + m\, \varphi) + B^\mu B_\mu + B^\mu \left(\partial^\nu B_{\nu\mu} - \partial_\mu \varphi + m\phi_\mu\right) - m^2 \bar{\beta}\beta$$

$$+ \left(\partial_\mu \bar{C}_\nu - \partial_\nu \bar{C}_\mu\right)\left(\partial^\mu C^\nu\right) - \left(\partial_\mu \bar{C} - m\bar{C}_\mu\right)\left(\partial^\mu C - mC^\mu\right) + \partial_\mu \bar{\beta}\, \partial^\mu \beta$$

$$+ (\partial \cdot \bar{C} + \rho + m\, \bar{C})\lambda + (\partial \cdot C - \lambda + m\, C)\rho, \tag{A1}$$

$$\mathcal{L}_{\bar{b}} = \frac{1}{12} H_{\mu\nu\eta} H^{\mu\nu\eta} - \frac{1}{4} m^2 B_{\mu\nu} B^{\mu\nu} - \frac{1}{4} \Phi_{\mu\nu} \Phi^{\mu\nu} + \frac{1}{2} m B_{\mu\nu} \Phi^{\mu\nu} - \bar{B}^2$$

$$+ \bar{B}(\partial \cdot \phi - m\,\varphi) + \bar{B}_\mu \bar{B}^\mu + \bar{B}^\mu \left(\partial^\nu B_{\nu\mu} + \partial_\mu \varphi + m\phi_\mu\right) - m^2\,\bar{\beta}\beta$$

$$+ \left(\partial_\mu \bar{C}_\nu - \partial_\nu \bar{C}_\mu\right)\left(\partial^\mu C^\nu\right) - \left(\partial_\mu \bar{C} - m\bar{C}_\mu\right)\left(\partial^\mu C - mC^\mu\right) + \partial_\mu \bar{\beta}\,\partial^\mu \beta$$

$$+ (\partial \cdot \bar{C} + \rho + m\,\bar{C})\lambda + (\partial \cdot C - \lambda + m\,C)\rho, \tag{A2}$$

where $\Phi_{\mu\nu} = \partial_\mu \phi_\nu - \partial_\nu \phi_\mu$ is the field-strength tensor for the *vector* Stückelberg field $\phi_\mu$. There are additional fermionic (anti-)ghost fields ($\bar{C}_\mu$, $C_\mu$, $\bar{C}$, $C$) and bosonic (anti-)ghost fields ($\bar{\beta}$)$\beta$ in our theory along with an additional scalar field $\varphi$ with ghost numbers $(-1) + 1, (-2) + 2$ and zero, respectively. There are a couple of additional Nakanishi–Lautrup-type fields ($\bar{B}$)$B$, too. It will be noted that the massive Abelian two-form field has the rest mass $m$ which happens to be the mass of the rest of the fields of our theory [31]. The above Lagrangian densities $\mathcal{L}_{\bar{b}}$ and $\mathcal{L}_b$ respect the following off-shell nilpotent $[s_{(a)b}^2 = 0]$ (anti-)BRST symmetry transformations $[s_{(a)b}]$, namely:

$$s_{ab}B_{\mu\nu} = -(\partial_\mu \bar{C}_\nu - \partial_\nu \bar{C}_\mu), \qquad s_{ab}\bar{C}_\mu = -\partial_\mu \bar{\beta}, \qquad s_{ab}\phi_\mu = \partial_\mu \bar{C} - m\,\bar{C}_\mu,$$

$$s_{ab}C_\mu = \bar{B}_\mu, \quad s_{ab}\beta = -\lambda, \quad s_{ab}\bar{C} = -m\,\bar{\beta}, \quad s_{ab}C = \bar{B}, \quad s_{ab}B = -m\,\rho,$$

$$s_{ab}\varphi = \rho, \qquad s_{ab}B_\mu = \partial_\mu \rho, \qquad s_{ab}[\bar{B}, \rho, \lambda, \bar{\beta}, \bar{B}_\mu, H_{\mu\nu\kappa}] = 0,$$

$$s_b B_{\mu\nu} = -(\partial_\mu C_\nu - \partial_\nu C_\mu), \qquad s_b C_\mu = -\partial_\mu \beta, \qquad s_b \phi_\mu = \partial_\mu C - m\,C_\mu,$$

$$s_b \bar{C}_\mu = -B_\mu, \quad s_b \bar{\beta} = -\rho, \quad s_b C = -m\,\beta, \quad s_b \bar{C} = B, \quad s_b \bar{B} = -m\,\lambda,$$

$$s_b \varphi = \lambda, \qquad s_b \bar{B}_\mu = -\partial_\mu \lambda, \qquad s_b[B, \rho, \lambda, \beta, B_\mu, H_{\mu\nu\kappa}] = 0. \tag{A3}$$

The above (anti-)BRST symmetries are *perfect* symmetries for the Lagrangian densities $\mathcal{L}_{\bar{b}}$ and $\mathcal{L}_b$, respectively, because we observe that:

$$s_{ab}\mathcal{L}_{\bar{b}} = \partial_\mu \left[\left(\partial^\mu \bar{C} + m\,\bar{C}^\mu\right)\bar{B} - \left(\partial^\mu \bar{C}^\nu - \partial^\nu \bar{C}^\mu\right)\bar{B}_\nu - \lambda\,\partial^\mu \bar{\beta} + \rho\,\bar{B}^\mu\right],$$

$$s_b \mathcal{L}_b = -\partial_\mu \left[\left(\partial^\mu C - m\,C^\mu\right)B + \left(\partial^\mu C^\nu - \partial^\nu C^\mu\right)B_\nu + \rho\,\partial^\mu \beta + \lambda\,B^\mu\right]. \tag{A4}$$

The above results establish that the action integrals $S_1 = \int d^D x\,\mathcal{L}_{\bar{b}}$ and $S_2 = \int d^D x\,\mathcal{L}_b$ remain invariant under the (anti-)BRST symmetry transformations $[s_{(a)b}]$, respectively, due to Gauss's divergence theorem.

There are a few comments in order here. First, we note that the kinetic term for the gauge field ($B_{\mu\nu}$), owing its origin to the exterior derivative (i.e., $H^{(3)} = d\,B^{(2)}$), remains invariant under the (anti-)BRST symmetry transformations $[s_{(a)b}]$. Second, we differ in some of our notations, a few terms in the coupled (but equivalent) Lagrangian densities and signs of the symmetry transformations from the earlier works [27,31]. Finally, we note that the (anti-)BRST symmetry transformations $[s_{(a)b}]$ are off-shell nilpotent $[s_{(a)b}^2 = 0]$ of order two and absolutely anticommuting (i.e., $\{s_b, s_{ab}\} = 0$) in nature, namely:

$$\{s_b, s_{ab}\}B_{\mu\nu} = \partial_\mu(B_\nu - \bar{B}_\nu) - \partial_\nu(B_\mu - \bar{B}_\mu),$$

$$\{s_b, s_{ab}\}\Phi_\mu = \partial_\mu(B + \bar{B}) - m(B_\mu - \bar{B}_\mu), \tag{A5}$$

provided we invoke the (anti-)BRST invariant

$$s_{(a)b}\left[B_\mu - \bar{B}_\mu - \partial_\mu \varphi\right] = 0, \qquad s_{(a)b}\left[B + \bar{B} + m\,\varphi\right] = 0, \tag{A6}$$

CF-type restrictions of our theory for the proof of the absolute anticommutativity in (A5).

The Noether conserved (anti-)BRST charges have been computed in [31]. We quote here these expressions (with appropriate and correct signs) as (see, e.g., [31] for details):

$$Q_{ab} = \int d^{D-1}x\left[-H^{0ij}\left(\partial_i \bar{C}_j\right) - \left(\partial^0 \bar{C}^i - \partial^i \bar{C}^0\right)\bar{B}_i + \left(\partial^0 \bar{C} - m\,\bar{C}^0\right)\bar{B}\right.$$

$$- \left( \partial_i \bar{C} - m\, \bar{C}_i \right) \left( \Phi^{0i} - m\, B^{0i} \right) + m \left( \partial^0 C - m\, C^0 \right) \bar{\beta}$$

$$- \left( \partial^0 C^i - \partial^i C^0 \right) \left( \partial_i \bar{\beta} \right) - \lambda\, \partial^0 \bar{\beta} + \rho\, \bar{B}^0 \Big],$$

$$Q_b = \int d^{D-1}x \Big[ - H^{0ij} \left( \partial_i C_j \right) - \left( \partial^0 C^i - \partial^i C^0 \right) B_i - \left( \partial^0 C - m\, C^0 \right) B$$

$$- \left( \Phi^{0i} - m\, B^{0i} \right) \left( \partial_i C - m\, C_i \right) - m \left( \partial^0 \bar{C} - m\, \bar{C}^0 \right) \beta$$

$$+ \left( \partial^0 \bar{C}^i - \partial^i \bar{C}^0 \right) \left( \partial_i \beta \right) - \rho\, \partial^0 \beta - \lambda\, B^0 \Big]. \tag{A7}$$

These Noether conserved charges are the generators for all the (anti-)BRST symmetry transformations (A3) which have been explicitly quoted in (A3). At this stage, we observe that the above Noether conserved (anti-)BRST charges are non-nilpotent. This can be checked explicitly by applying the BRST symmetry transformations ($s_b$) on $Q_b$ and anti-BRST symmetry transformation ($s_{ab}$) on $Q_{ab}$ as illustrated below:

$$s_b Q_b = -i \left\{ Q_b, Q_b \right\} = - \int d^{D-1}x \left[ m \left( \partial^0 B + m\, B^0 \right) \beta + \left( \partial^0 B^i - \partial^i B^0 \right) \partial_i \beta \right] \neq 0,$$

$$s_{ab} Q_{ab} = -i \left\{ Q_{ab}, Q_{ab} \right\} = \int d^{D-1}x \left[ m \left( \partial^0 \bar{B} - m\, \bar{B}^0 \right) \bar{\beta} - \left( \partial^0 \bar{B}^i - \partial^i \bar{B}^0 \right) \partial_i \bar{\beta} \right] \neq 0. \tag{A8}$$

Thus, we note that the Noether (anti-)BRST charges (i) are conserved quantities, (ii) are the generators [28] for the appropriate (anti-)BRST symmetry transformations, and (iii) are found to be *not* nilpotent of order two (i.e., $Q_b^2 \neq 0$, $Q_{ab}^2 \neq 0$).

In what follows, we follow a systematic method to convert the non-nilpotent Noether conserved (anti-)BRST charges (A7) into their off-shell nilpotent versions. Since we have demonstrated our method in the context of the BRST charge for the Stückelberg-modified massive Abelian three-form theory (cf. Section 5), we derive the off-shell nilpotent version $[Q_{ab}^{(1)}]$ of the non-nilpotent Noether anti-BRST $Q_{ab}$ following our proposal. First of all, we apply the Gauss divergence theorem whch turns the *first* term of $Q_{ab}$ (cf. Equation (A7)) as

$$\left( \partial_i H^{0ij} \right) \bar{C}_j + m \left( \Phi^{0i} - m\, B^{0i} \right) \bar{C}_i = \left( \partial^0 \bar{B}^i - \partial^i \bar{B}^0 \right) \bar{C}_i, \tag{A9}$$

where we have used the following EL-EoM that is derived from $\mathcal{L}_{\bar{b}}$

$$\partial_\mu H^{\mu\nu\lambda} + \left( \partial^\nu \bar{B}^\lambda - \partial^\nu \bar{B}^\lambda \right) - m \left( \Phi^{\nu\lambda} - m\, B^{\nu\lambda} \right) = 0. \tag{A10}$$

We claim that the r.h.s. of (A9) will be present in the off-shell nilpotent version of the anti-BRST charge $[Q_{ab}^{(1)}]$.

Now, let us focus on the remaining part of the *fourth* term inside the integration. We note that using the Gauss divergence theorem, we obtain

$$- \left( \Phi^{0i} - m\, B^{0i} \right) \partial_i \bar{C} \equiv \partial_i \left[ \left( \Phi^{0i} - m\, B^{0i} \right) \bar{C} \right]. \tag{A11}$$

At this stage, we use the following equation of motion in the above:

$$\partial_\mu \Phi^{\mu\nu} - \partial^\nu \bar{B} - m \left( \partial_\mu B^{\mu\nu} \right) + m\, \bar{B}^\nu = 0. \tag{A12}$$

The appropriate substitution leads to the following

$$\partial_i \left[ \left( \Phi^{0i} - m\, B^{0i} \right) \bar{C} \right] = - \dot{\bar{B}}\, \bar{C} + m\, \bar{B}^0\, \bar{C} \equiv - \left( \dot{\bar{B}} - m\, \bar{B}^0 \right) \bar{C}. \tag{A13}$$

Ultimately, the sum of the *first* term and *fourth* term (i.e., sum of (A10) and (A13)) leads to the following two terms:

$$+ \left( \partial^0 \bar{B}^i - \partial^i \bar{B}^0 \right) \bar{C}_i - \left( \dot{\bar{B}} - m\, \bar{B}^0 \right) \bar{C}. \tag{A14}$$

These two terms will stay in the off-shell nilpotent version of the anti-BRST charge $[Q_{ab}^{(1)}]$. If we apply an anti-BRST symmetry transformations $s_{ab}$ on (A14), we obtain the following

$$s_{ab}\left[(\partial^0 \bar{B}^i - \partial^i \bar{B}^0)\, \bar{C}_i - (\dot{\bar{B}} - m\, \bar{B}^0)\, \bar{C}\right] = -(\partial^0 \bar{B}^i - \partial^i \bar{B}^0)\, \partial_i\, \bar{\beta} + m\, (\dot{\bar{B}} - m\, \bar{B}^0)\, \bar{\beta}. \quad \text{(A15)}$$

We have to look, according to our proposal, and see whether the above terms in (A15) can cancel and/or can be added and/or subtracted with any terms of $Q_{ab}$ (i.e., Noether charge). We find that the following modifications of the *fifth* and *sixth* terms, namely:

$$+m\, (\partial^0 C - m\, C^0)\, \bar{\beta} - (\partial^0 C^i - \partial^i C^0)\, \partial_i\, \bar{\beta} \equiv 2\, m\, (\partial^0 C - m\, C^0)\, \bar{\beta},$$

$$-2\, (\partial^0 C^i - \partial^i C^0)\, \partial_i\, \bar{\beta} - m\, (\partial^0 C - m\, C^0)\, \bar{\beta} + (\partial^0 C^i - \partial^i C^0)\, \partial_i\, \bar{\beta}, \quad \text{(A16)}$$

will enable us to observe that the anti-BRST symmetry transformations $s_{ab}$ on the last *two* terms of the above equation, namely:

$$s_{ab}\left[(\partial^0 C^i - \partial^i C^0)\, \partial_i\bar{\beta} - m\, (\partial^0 C - m\, C^0)\, \bar{\beta}\right] = (\partial^0 \bar{B}^i - \partial^i \bar{B}^0)\, \partial_i\bar{\beta} - m\, (\dot{\bar{B}} - m\, \bar{B}^0)\, \bar{\beta}, \quad \text{(A17)}$$

cancels out with the ones that have been obtained in (A15). Hence, we note that in addition to the terms in (A14), the terms in the square bracket of (A17) will also stay in the off-shell nilpotent version $[Q_{ab}^{(1)}]$ of the anti-BRST charge.

We concentrate now on the leftover terms of (A16), which are as follows

$$2\, m\, (\partial^0 C - m\, C^0)\, \bar{\beta} - 2\, (\partial^0 C^i - \partial^i C^0)\, \partial_i\bar{\beta}. \quad \text{(A18)}$$

Using the Gauss divergence theorem, we note that the last term can be written as

$$2\, \partial_i (\partial^0 C^i - \partial^i C^0)\, \bar{\beta}, \quad \text{(A19)}$$

because both the terms of (A18) are inside the integration. At this stage, we use the following EL-EoM derived from $\mathcal{L}_{\bar{b}}$, namely:

$$\partial_\mu (\partial^\mu C^\nu - \partial^\nu C^\mu) + \partial^\nu \lambda - m\, (\partial^\nu C - m\, C^\nu) = 0, \quad \text{(A20)}$$

which leads to the following for the choice $\nu = 0$, namely:

$$\partial_i (\partial^0 C^i - \partial^i C^0)\, \bar{\beta} = 2\, \dot{\lambda}\, \bar{\beta} - 2\, m\, (\dot{C} - m\, C^0)\, \bar{\beta}. \quad \text{(A21)}$$

The above relationship plays an important and decisive role. For instance, the substitution of the above into equation (A18) leads to

$$2\, m\, (\dot{C} - m\, C^0)\, \bar{\beta} + 2\, \partial_i\, (\partial^0 C^i - \partial^i C^0)\, \bar{\beta} = 2\, \dot{\lambda}\, \bar{\beta}. \quad \text{(A22)}$$

We find that if we apply an anti-BRST symmetry transformation $(s_{ab})$ on the r.h.s. of (A22), it turns out to be zero. Thus, the theoretical tricks of our proposal *terminate* here. Hence, the term $(2\, \dot{\lambda}\, \bar{\beta})$ will stay in the off-shell nilpotent version of the $[Q_{ab}^{(1)}]$ of the anti-BRST charge. Finally, we have the off-shell nilpotent version of the anti-BRST charge $[Q_{ab}^{(1)}]$ (derived from the non-nilpotent Noether conserved anti-BRST charge) as follows

$$Q_{ab} \longrightarrow Q_{ab}^{(1)} = \int d^{D-1}x \left[(\partial^0 \bar{B}^i - \partial^i \bar{B}^0)\, \bar{C}_i - (\bar{B}^0 - m\, \bar{B}^0)\, \bar{C} + (\partial^0 C^i - \partial^i C^0)\, \partial_i\bar{\beta} \right.$$

$$\left. - m\, (\dot{C} - m\, C^0)\, \bar{\beta} + 2\, \dot{\lambda}\, \bar{\beta} - \lambda\, \dot{\bar{\beta}} + \rho\, \bar{B}^0 + (\dot{\bar{C}} - m\, \bar{C}^0)\, \bar{B} - (\partial^0 \bar{C}^i - \partial^i \bar{C}^0)\, \bar{B}_i\right], \quad \text{(A23)}$$

where we note that some of the original terms of the Noether conserved anti-BRST charge $Q_{ab}$ are present, which are found to be the anti-BRST invariant quantities, namely:

$$s_{ab}(\lambda \, \dot{\bar{\beta}}) = 0, \quad s_{ab}(\rho \, \bar{B}^0), \quad s_{ab}[(\dot{\bar{C}} - m \, \bar{C}^0) \, \bar{B}] = 0, \quad s_{ab}[(\partial^0 \bar{C}^i - \partial^i \bar{C}^0) \, \bar{B}_i] = 0. \quad \text{(A24)}$$

It is now straightforward to check that the following is true, namely:

$$s_{ab} Q_{ab}^{(1)} = -i \, \{Q_{ab}^{(1)}, Q_{ab}^{(1)}\} = 0 \quad \Longrightarrow \quad [Q_{ab}^{(1)}]^2 = 0, \quad \text{(A25)}$$

where the l.h.s. can be explicitly computed using the application of the anti-BRST symmetry transformation $(s_{ab})$ on the explicit expression (A23).

We end this appendix with the final remark that following the theoretical tricks of our proposal, it is a bit of involved but straightforward algebra that leads to the deduction of the off-shell nilpotent $([Q_b^{(1)}]^2 = 0)$ version of the BRST charge $[Q_b^{(1)}]$ from the non-nilpotent Noether BRST charge $(Q_b)$ as follows:

$$Q_b \longrightarrow Q_b^{(1)} = \int d^{D-1}x \left[ (\partial^0 B^i - \partial^i B^0) \, C_i + (\dot{B} + m \, B^0) \, C + m \, (\dot{\bar{C}}^0 - m \, \bar{C}^0) \, \beta \right.$$

$$\left. - (\partial^0 \bar{C}^i - \partial^i \bar{C}^0) \, \partial_i \beta + 2 \dot{\rho} \, \beta - \rho \, \dot{\beta} - \lambda \, B^0 - (\partial^0 C^i - \partial^i C^0) \, B_i - (\dot{C} - m \, C^0) \, B \right]. \quad \text{(A26)}$$

It is an elementary exercise to check that we have

$$s_b Q_b^{(1)} = -i \, \{Q_b^{(1)}, Q_b^{(1)}\} = 0 \quad \Longrightarrow \quad [Q_b^{(1)}]^2 = 0, \quad \text{(A27)}$$

where the l.h.s. is computed by an explicit application of $s_b$ on the BRST charge $[Q_b^{(1)}]$ (cf. Equation (A26)) which establishes that we have obtained the off-shell nilpotent version of the BRST charge $[Q_b^{(1)}]$ from the non-nilpotent conserved Noether BRST charge $(Q_b)$.

## Notes

1.    It is found that the Noether conserved (anti-)BRST charges are the generators of the off-shell nilpotent (anti-)BRST symmetry transformations from which they are derived by using the Noether theorem.

2.    The higher $p$-form $(p = 2, 3, \dots)$ gauge fields (and corresponding theories) are important because such fields appear in the quantum excitations of the (super)string theories (see, e.g., [14] for details).

3.    The (anti-)BRST symmetry transformations (4) and (5) are absolutely anticommuting (i.e., $\{s_b, s_{ab}\} = 0$) in nature on the submanifold of the Hilbert space of *quantum* variables, which is defined by the (anti-) BRST invariant (i.e., $s_{(a)b} [b + \bar{b} + 2 \bar{\beta} \beta] = 0$) CF-type restriction: $b + \bar{b} + 2 \bar{\beta} \beta = 0$.

4.    The off-shell nilpotent $[s_{(a)b}^2 = 0]$ (anti-)BRST symmetry transformations $s_{(a)b}$ (cf. Equations (4) and (5)) are said to be *perfect* symmetry transformations for the Lagrangians $L_{(\bar{b})b}$, respectively, because of our observations in (6) and (7) where *no* EL-EoMs and/or CF-type restriction(s) are *used* for their validity.

5.    This is due to the fact that, as pointed out earlier, we wish to obtain $Q_b^{(1)}$ such that $s_b \, Q_b^{(1)} = 0$.

6.    It will be noted that, in the cases of the 1D massive spinning relativistic particle and D-dimensional non-Abelian 1-form gauge theory, the *original* term of the Noether conserved charges are good enough to ensure that: $s_b \, Q_b^{(1)} = 0$, $s_{ab} \, Q_{ab}^{(1)} = 0$, $s_b \, Q_B^{(1)} = 0$ and $s_{ab} \, Q_{\bar{B}}^{(1)} = 0$.

7.    The Lagrangian densities $\mathcal{L}_B$ and $\mathcal{L}_{\bar{B}}$ are coupled and equivalent due to the existence of *six* (anti-)BRST invariant CF-type restrictions on our theory which have been *systematically* derived using the augmented version of superfield approach to BRST formalism in [25].

8.    It is worthwhile to point out here that the number of fields and corresponding equations of motion become too many even in the case of the modified version of a massive Abelian three-form gauge theory (see. e.g., [26,27] for details), and it becomes very difficult to know which equation of motion will be picked up from amongst the *total* number of EL-EoMs, to render the non-nilpotent versions of the Noether conserved charges into the off-shell *nilpotent* versions of the conserved charges. Moreover, the expressions for the conserved Noether charges themselves become very complicated and cumbersome (see, e.g., [26] for details) when we discuss the modified massive higher $p$-form $(p = 2, 3, \dots)$ gauge theories.

[9] There are *two* ways by which $s_b \, Q_b^{(1)} = 0$ and $s_{ab} \, Q_{ab}^{(1)} = 0$ can be proven because of the integration present in the expressions for $Q_{(a)b}^{(1)}$. The first option is a clear-cut proof that the *total* integrand turns out to be *zero* when we apply $s_{(a)b}$ on it. The second option is the case when the integrand transforms to a *total space* derivative under the applications of $s_{(a)b}$. In our present endeavor, it is the first option that has been chosen where we have proven that $s_b \, Q_b^{(1)} = 0$ and $s_{ab} \, Q_{ab}^{(1)} = 0$.

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
