# Peer review of "Noether Theorem and Nilpotency Property of the (Anti-)BRST Charges in the BRST Formalism: A Brief Review"

_universe, doi:10.3390/universe8110566_

Round 1
Reviewer 1 Report
See pdf file

Author Response
We attatch herewith the PDF file of our reply to our esteemed first Reviewer.

Reviewer 2 Report
The paper is devoted to the investigation of the (anti)-BRST transformations and (anti)-BRST charges in some theories invariant under certain gauge transformations. The authors claim that the (anti)-BRST charges constructed with the help of the Noether theorem are not nilpotent off shell despite the fact that they generate the nilpotent transformations. They also derive the corresponding nilpotent charges using a certain procedure. However, the statement that the Noether BRST charge is not nilpotent seems for me rather unexpectable and weird. That is why I believe that the paper can be published only if the authors find a clear reason of this unusual fact and describe it in all necessary details. Also I would like to note that the Minkowski metric on the page 4 is written incorrectly and needs to be corrected.
Author Response
We attach herewith the PDF file of our esteemed second Reviewer.

Reviewer 3 Report
In this review article, the authors examined the BRST charges of certain gauge theories with Curci-Ferrari-type gauge fixing conditions. They found that even though they started with off-shell nilpotent BRST transformations, the resultant BRST charges obtained as Noether charges of the corresponding transformation, fail to be off-shell nilpotent. The authors also proposed a systematic recipe to amend it and presented how to obtain off-shell nilpotent BRST charges.
Overall, this article deals with a fundamental property of gauge theory in a specific gauge choice and serves interesting observations. The details are presented clearly and sufficient examples are included. Thus, its scientific merit is sufficient.
I then want to ask the authors to explain more about the physical significance of the off-shell nilpotency.
The nilpotency of the BRST charge is crucial when we define a physical subspace of the Hilbert space of a given gauge theory as BRST cohomology. The original BRST charges defined through the Noether procedure are inappropriate for this purpose? Or, it is sufficient but using the extended version (with off-shell nilpotency) makes the physical structure transparent? This kind of discussion should be included. (Namely, the requirement of off-shell nilpotency is not only mathematical curiosity but has physical significance).
As for the presentation, I would raise difficulty over the language. There is an overuse of emphases by using the italic style, which reduces the readability of the article quite a bit. I am confused about which parts are important or stressed. The authors should reduce them and make emphasis truly important phrases.
With these points amended, I would recommend the paper for publication in Universe.
Author Response
We attach herewith the PDF file of our esteemed third Reviewer.

Round 2
Reviewer 1 Report
I have gone through the revised version of the manuscript and the authors' response. The authors have improved their manuscript and addressed all the issues mentioned in the previous report sufficiently. In conclusion, I recommend publication of the revised manuscript.
Author Response
We have attached our reply to our esteemed first Reviewer.

Reviewer 2 Report
In the new version of the paper I did not find an explanation of the reason why the Noether (anti)-BRST charge can be non-nilpotent (although it seems for me that the calculations are correct). If we calculate the square of the BRST transformation, then for an arbitrary field \varphi
0 = \delta_1 \delta_2 \varphi = [\varepsilon_1 Q, [\varepsilon_2 Q, \varphi]]
= - \varepislon_1\varepsilon_2 [Q,[Q,\varphi\}\} = -\varepsilon_1 \varepsilon_2 [Q^2,\varphi]
due to Z_2 graded Jacoby identity. Then one is tempted to set Q^2=0. However, the authors claim that this is not true. Possibly, one may suggest that [Q^2,\varphi] is proportional to the motion equations, but the authors do not explain this properly. I believe that this question should be clearly answered in the paper. Also, I recommend to correct a misprint in the third line after eq. (67) and cite the original Becchi, Rouet, Stora, and Tyutin papers.
Author Response
We have attached the PDF file of our reply to our esteemed second Reviewer.
